# Creep-dilatancy development at a transform plate boundary

Nabil Sultan [1✉], Shane Murphy[1], Vincent Riboulot [1] & Louis Géli [1]

How tectonic plates slip slowly and episodically along their boundaries, is a major, open question in earthquake science. Here, we use offshore in-situ sediment pore-pressure acquired in the proximity of the active offshore Main Marmara Fault and onshore geodetic time-series data set from a single GPS station to demonstrate the pore-pressure/deformation coupling during a 10-month slow-slip event. We show that pore pressure fluctuations are the expression of hydro-mechanical process affecting the deep seismogenic zone and indicate that small disturbances in geodetic data may have important meaning in terms of transient deformations. These results have major implications in understanding the spatial impact of slow-slip processes and their role in earthquake cycles. We demonstrate that piezometers measuring along a transform fault can help define the time scale regulating the coupling between slow-slip events and earthquake nucleation process.

---

[1] Geo-Ocean UMR6538, Ifremer, CNRS, UBO, UBS, 29280 Plouzané, France. ✉email: nabil.sultan@ifremer.fr

That tectonic plates may slip slowly and episodically along segments of their boundaries is one of the most intriguing, unexplained observations in solid earth geophysics, as it reveals a continuum of transient deformation along active faults, ranging from seismic rupture to aseismic events[1,2]. While the major role and influence of pore pressure has long been proposed[3–5], it has never been observed because of the difficulty in linking deep processes at seismogenic depths with near-surface pore-pressure measurements. Slow slip events (SSEs) have mostly been studied using on-shore Global Positioning System (GPS) data[6], where the discovery of fluid flow based on direct, in situ observations from the seafloor along the subducting plate interfaces have led to the hypothesis of a causal relationship between SSEs and changes in fluid activities at the fault zone[7–9].

The Sea of Marmara—south of Istanbul—has become, after 20 years of investigations, one of the most well-known submarine domains on earth. Studies have now largely converged to a scheme with a master fault—the Main Marmara Fault[10] (MMF; red line in Fig. 1A)—taking 75% of the right-lateral geodetic slip rate (~20 mm/year) between the Eurasian Plate and the Anatolian block[10–13]. The western segments of the MMF are characterized by a higher level of background seismicity compared to the eastern segments. The common occurrence of small repeating earthquakes[14] suggests that the MMF has a deep creep component in the western Sea of Marmara, but not in the central nor in the eastern segments[15]. Based on submarine, acoustic ranging data, aseismic creep rates of 9–16 mm/year[16] were measured in the Western High area (WH in Fig. 1A), where active seeps[17–19] were found in association with an active mud volcano (MV) located <1 km away from the MMF trace (Fig. 1B, C). The largest gas emissions along the MMF were found to occur within a 900 m to 1000 m zone on either side of the MMF[20], which encompassed the MV and have been interpreted as being the trace of a high-permeability damage zone[21].

Here we analyze offshore in situ sediment pore-pressure acquired from the active MMF and onshore GPS time-series data. Our results indicate that a pore-pressure/deformation coupling exists during a 10-month slow-slip event. These observations demonstrate the role of dilatancy in regulating tectonic slip, reactivating an adjacent MV and far-field tectonic deformations.

## Results

In this context, two differential piezometers[22] measuring pore pressures in excess of hydrostatic pressures ("Methods") were deployed from Oct 2013 to Nov 2014[23] within the MMF valley

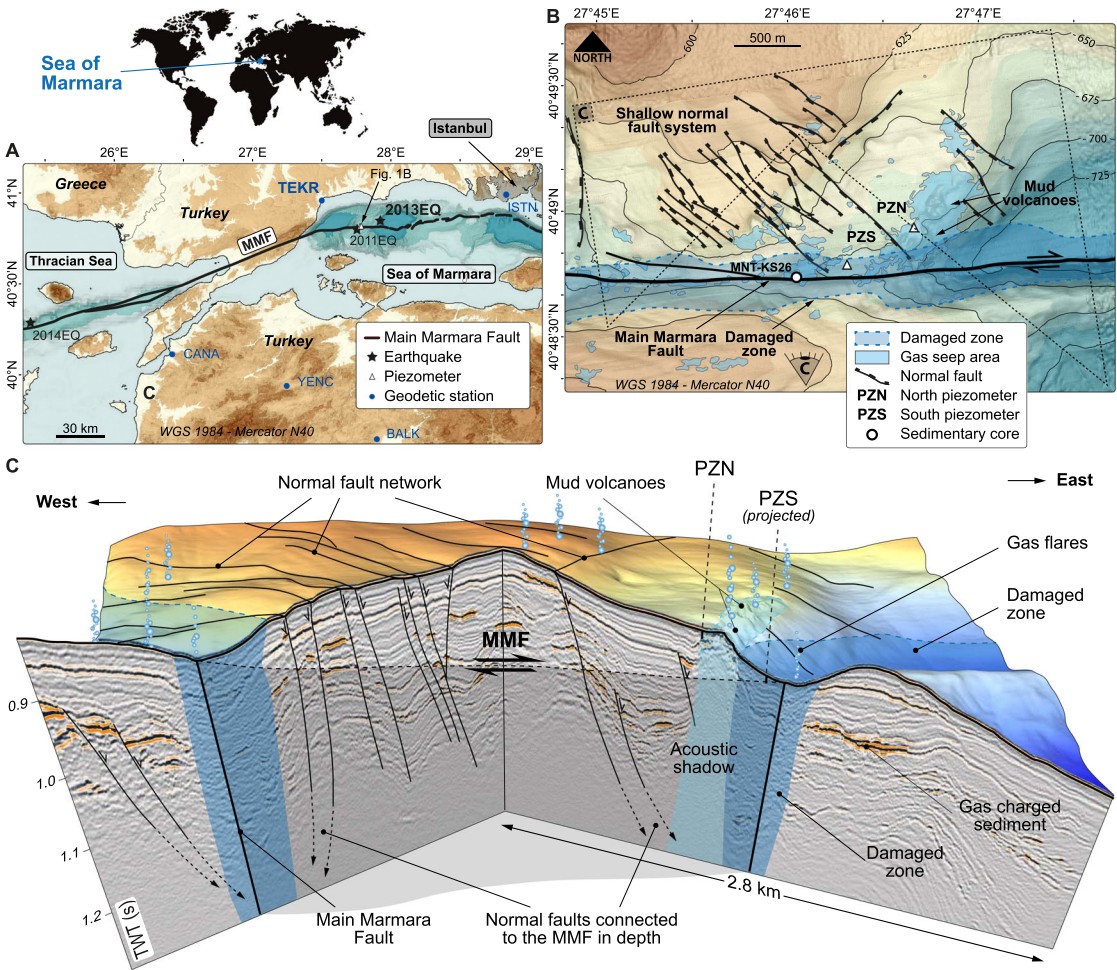

**Fig. 1 Study area. A** General map showing the study area, the onshore GPS stations (dots indicating from west to east: CANA, YENC, TEKR, BALK, and ISTN stations), and the 2011EQ, 2013EQ, and 2014EQ earthquake epicenters (black stars). The thick black line[11] indicates the trace of the northern strand of the North-Anatolian Fault. Background map boundaries are provided by ArcGIS Data and Maps. Redistribution rights are granted. https://www.esri.com/content/dam/esrisites/en-us/media/legal/redistribution-rights/redist-rights-2021.pdf. **B** Zoom of bathymetry showing piezometers PZN and PZS and their positions with respect to the Main Marmara Fault (MMF) and its damaged zone (Supplementary Fig. 1), the mud volcanoes (MVs), and gas seep areas[17]. **C** 3D diagram including interpreted seismic lines extracted from 3D HR seismic data and indicating the position of PZN and PZS.

and close to the MV, respectively (PZS and PZN in Fig. 1), to investigate the coupling between the hydrogeological and tectonic systems along the WH segment of the MMF. The PZS and PZN water depths are 667 and 672 m, respectively.

During the monitoring period, two notable earthquakes occurred at epicentral distances from PZN of 12.5[24] and 209 km[25], respectively: the 11 km deep, Mw 4.8 27/11/2013 (2013EQ) nucleated along the MMF in the area where repeating earthquakes have been documented, while the 9 km deep, Mw 6.9 24/5/2014 earthquake (2014EQ) struck the Ganos segment of the NAF in the Thracian Sea (epicenters in Fig. 1A).

In parallel, we analyze 7-year period GPS data[26] using publicly available data from Nevada Geodetic Laboratory-NGL (http://geodesy.unr.edu/index.php). GPS time series data are provided in the IGS14[27] reference frame and in the plate fixed reference frame by considering the plate on which the station lies. For this last case, the plate motion trends are removed for the horizontal component time series. The accuracy of an observed position in the north–south, east–west, and in up–down directions depends significantly on the duration of the observing period and on the accuracy of the GPS satellite orbits used during processing[28]. In a previous study, an accuracy analysis of relative positions of permanent GPS stations in the Marmara Region has shown that the root mean square error is within 1 mm for the north–south, east–west components while it is between 2 and 3 mm for the up–down direction[16]. For the present study, we used the STL (seasonal-trend decomposition based on LOESS)[29] procedure in R[30], which is a filtering procedure that decomposes a time series into three components: trend, seasonal, and remainder parts. Figure 2 shows a decomposition plot of TEKR geodetic data with a trend of the northern displacement (with respect to fixed EU plate) strongly affected by the 2013EQ. The vertical displacement at TEKR also indicates a perturbation of the trend between 2013EQ and 2014EQ.

The horizontal displacement (north with respect to fixed EU plate)[6], recorded by the four westernmost stations, indicates a

general evolution to the south in agreement with the geodetic strain rates within the region[31] (Fig. 3A). The non-corrected vertical (up) GPS displacement indicates seasonal variations (Fig. 3A). Superimposed on this general trend, displacements (north and up) at station TEKR seem disturbed (arrows in Figs. 2 and 3) during a period following the 2013EQ. The eastern horizontal GPS displacement from the Anatolian plate evolves smoothly to the west independently of 2013EQ (Supplementary Fig. 2).

Excess pore-pressure data (i.e., above hydrostatic) measured at 7.08 mbsf (PZN-P6) and 6.28 mbsf (PZS-P5) show two opposite trends (Fig. 3B and Supplementary Figs. 3–6). The PZN-P6 pressure increases over a period greater than 3.5 months starting around 1 week after the 2013EQ. Pore-pressure and temperature data from PZN-P6 and PZN-P5 suggest a diffusion–advection process controlling the thermal and pressure gradients (Supplementary Figs. 7, 8, and 10). During this period, the pore pressure repeatedly reaches hydro-fracture conditions (Supplementary Fig. 11). After this, a smooth decay of the pressure is followed by a 3-month period of negative pore pressure (gray dashed area in Fig. 3B). The sensor positioned 0.8 m above P6 was the only one to record significant pressure changes during the monitoring period (Supplementary Fig. 3). At PZS-P5 station, negative pore pressure develops 1 week after the 2013EQ and lasts for 2.5 months (Fig. 3B). The P5 sensor was the only one to record significant pressure changes during deployment (Supplementary Fig. 5). The zoom diagrams in Fig. 3C show in detail the pore pressures generated by 2013EQ and 2014EQ during the monitoring period. The five other sensors of PZN and PZS were also disturbed by those two events (Supplementary Figs. 4–6).

**Near-field pore-pressure data versus far-field geodetic displacements.** Comparison in Fig. 4A between the Tekirdag daily mean temperatures (from climexp.knmi.nl) and the vertical displacement from TEKR station confirms the seasonal origin of the geodetic vertical component, where the redistributions of continental water mass deform elastically the Earth crust similar to

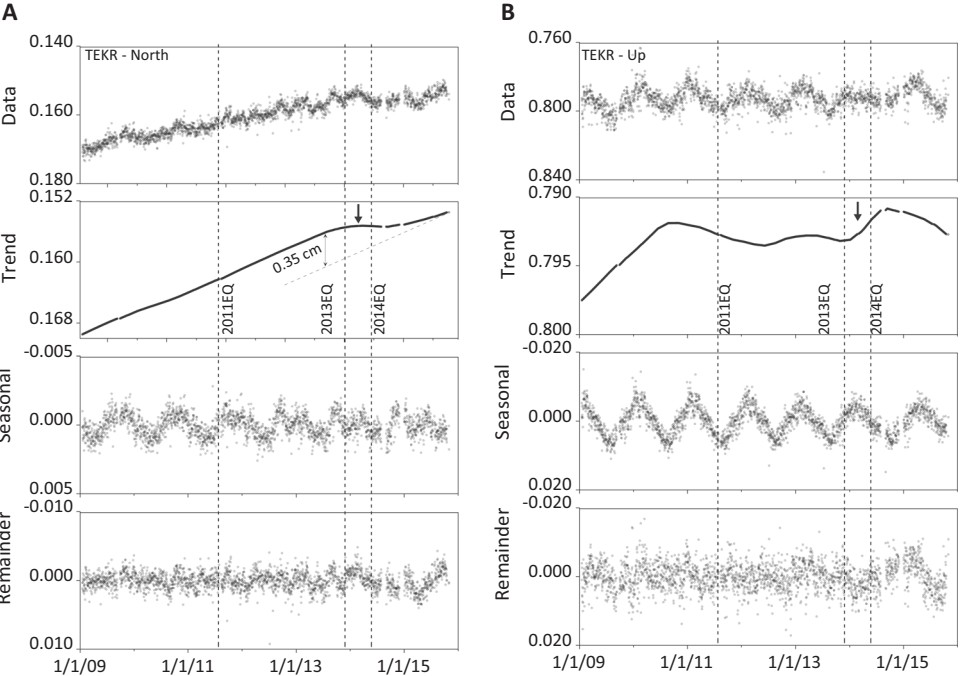

**Fig. 2 Decomposition plot of TEKR GPS geodetic data. A** Northern displacement and **B** vertical displacement. The unit on the vertical scale is meters. The horizontal scale uses the convention of day, month, and year. The timing of the three earthquakes (2011EQ, 2013EQ, and 2014EQ) are displayed by dashed vertical lines.

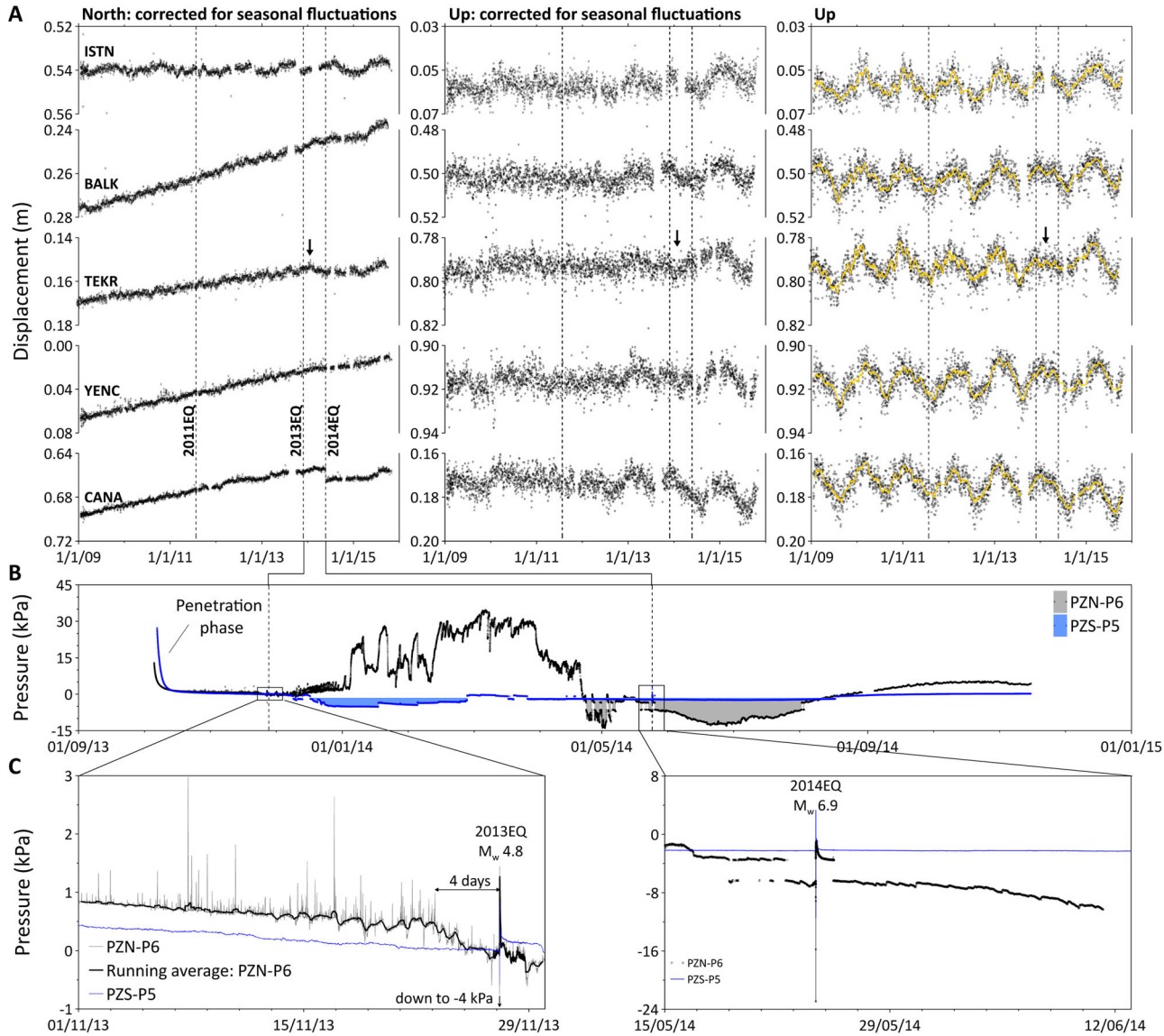

**Fig. 3 Time series data. A** Displacement time series (from left to right: north and up components corrected for seasonal fluctuations and non-corrected up component) for five sites surrounding the Sea of Marmara. From west to east: CANA, YENC, TEKR, BALK, and ISTN stations (locations in Fig. 1A). The three vertical dashed lines indicate the timing of three neighboring earthquakes (25 July 2011—2011EQ, 2013EQ, and 2014EQ). The range of the vertical axes is adjusted to better visualize the majority of the data. Arrows indicate perturbations that occur between 2013EQ and 2014EQ. **B** Pore-pressure time series measured at depths below the seabed of 7.08 m (PZN-P6) and 6.28 m (PZS-P5). **C** Enlargement diagrams showing the pore-pressure evolutions during 2013EQ and 2014EQ. The horizontal scale uses the convention of day, month, and year.

what was described previously[32–35]. Additionally, Fig. 4A highlights a 10-month period after the 2013EQ where the vertical displacement is almost locked with a displacement deficit of ±1 cm. Simultaneously and after the 2013EQ, pore pressure recorded by PZN-P6 synchronizes remarkably well with the TEKR GPS northern displacement indicating a strong link between the pore pressure from the MV (e.g., measured at PZN-P6) and the regional tectonic activity. The northern displacement shifts by around 0.35 cm to the north of the linear general trend that precedes 2013EQ (Fig. 2A).

Several authors have suggested, based on indirect evidence, a relationship between fluid activities and tectonic stresses[5,17,20,36]. Our data provide clear confirmation that regional strain (TEKR GPS data) and near-surface sediment pore pressure (PZN-P6) are interconnected (Fig. 4B). It is evident that a shallow localized process at the level of the piezometer is impossible to be detected

by onshore GPS data and vice versa. Therefore, the data suggest that shearing along the fault disturbs both the near-field sediment pore pressure and the far-field displacement.

The positive pore pressure recorded by PZN-P6 coincides with a displacement at TEKR toward the South (until 20/02/2014) while the pore-pressure decrease matches with a displacement to the North. In parallel, the locked vertical displacement at TEKR station during the wet period (high precipitations) indicates a net upward movement while the locked vertical displacement during the dry period implies a net downward vertical movement (Fig. 4A). The similarity between shallow pore water pressure and geodetic data suggests that pore-pressure perturbations have a physical and mechanical significance related to MMF at depth. We propose that this is due to the MV, which acts as a window to the MMF seismogenic zone linking stress/strain changes at depth to shallow pore-pressure variations.

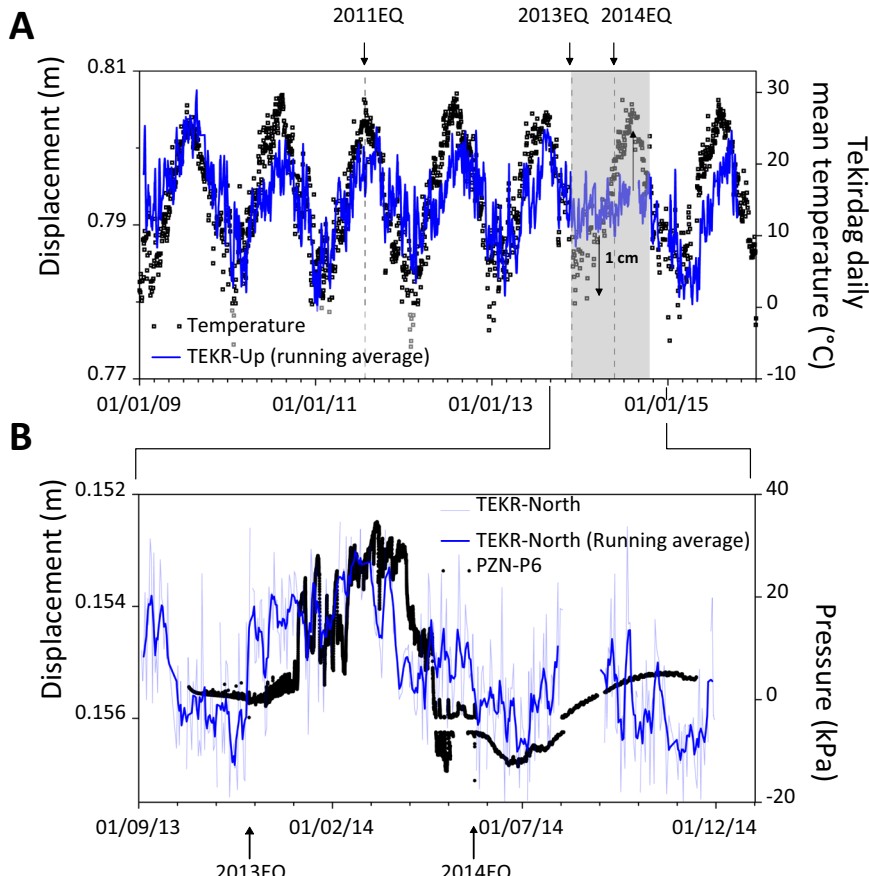

**Fig. 4 Pore-pressure versus geodetic data. A** Vertical displacement from TEKR geodetic station (non-corrected for seasonal fluctuations—blue line) compared to Tekirdag daily mean temperatures (black squares) suggesting a seasonal ground surface displacement except for a 10-month period that begins in November 2013 (gray area). **B** Pore pressure measured at 7.08 mbsf during a 13-month period at PZN station (PZN-P6) synchronize after the 2013EQ with the TEKR GPS northern displacement (TEKR-North) corrected for seasonal fluctuations.

**Negative pore pressure and dilatancy**. To illuminate the extent of the mechanical process controlling the near and far-field perturbations shown in Fig. 4, we hypothesize that the PZS-P5 negative pore pressure that developed only 1 week after the 2013EQ may be the result of an aseismic creeping activity along the MMF, which occurred in response to dilatancy due to shearing. Indeed, the PZS-P5 sensor is positioned within a silty–sandy layer (Supplementary Fig. 9) potentially dilatant during shearing. This shallow observed dilatancy is most probably the expression of a deeper and more generalized creeping process[16], which affects the MV pore pressure and the regional displacement field. In contrast, the other five sensors at PZS positioned within clayey sediments did not show any pore-pressure perturbation during the monitoring period (Supplementary Fig. 9).

**Slow slip along the offshore fault disturbing the onshore geodetic data**. To test the shear-dilatancy hypothesis, we use a numerical model to describe the SSE. Using a purely elastic half-space dislocation model[37], we determine how displacement along the MMF affects surface deformation at TERK. To do this, the MMF was discretized into rectangles no larger than 2 km by 2 km. The deformation caused by different dislocations source types (i.e., pure right-lateral slip and a mix of right-lateral and dilatant slip) at different depths and locations were calculated (Methods and Supplementary Figs. 12–16). In order to reproduce the same trend as that observed at TEKR, the best fit is obtained with a slipping surface comprising of a shallow slipping patch (i.e., <8 km)

containing both right-lateral slip and a transient inflation component that propagates at 0.5 km/day from east to west (Fig. 5A and case C in Supplementary Table 1). These best fitting scenarios of a shallow slipping event are consistent with results from the literature obtained from the same region[16,38,39]. The model, however, predicts an east–west positive displacement not detectable on the TEKR data (Supplementary Fig. 2) suggesting that an isotropic and homogeneous elastic three-dimensional (3D) model with uniform slip may not be able to reproduce all the complexity of a natural system. The reality is more complex with a spatial variable slip; for example, less slip along the eastern section of MMF relative to TEKR compared to the western section would possibly reduce the simulated positive east-west motion at TEKR. An alternative explanation could be the amplification of north-south motion by normal faults to the north of MMF. These scenarios would require more complex models and data for validation.

By correlating the pore-pressure data from PZS-P5 with the longitude position of slipping (Fig. 5B), it is found that the 50-day recorded dilatancy occurred during a slipping event along the WH segment of the MMF (S1 in Fig. 5C). The shearing dilatancy and associated pore pressure increase the normal effective stress leading to a more stable slipping event[40]. As a result, pore pressure slowly returns to equilibrium when the slow slip propagates to the west of WH. Proctor and co-authors[41] have measured pore-pressure transients in hydraulically isolated laboratory faults during seismic and SSEs. Those authors showed the importance of pore fluid–rock interactions, which may dominate and control both stability and failure time of faults. Our field data confirm the

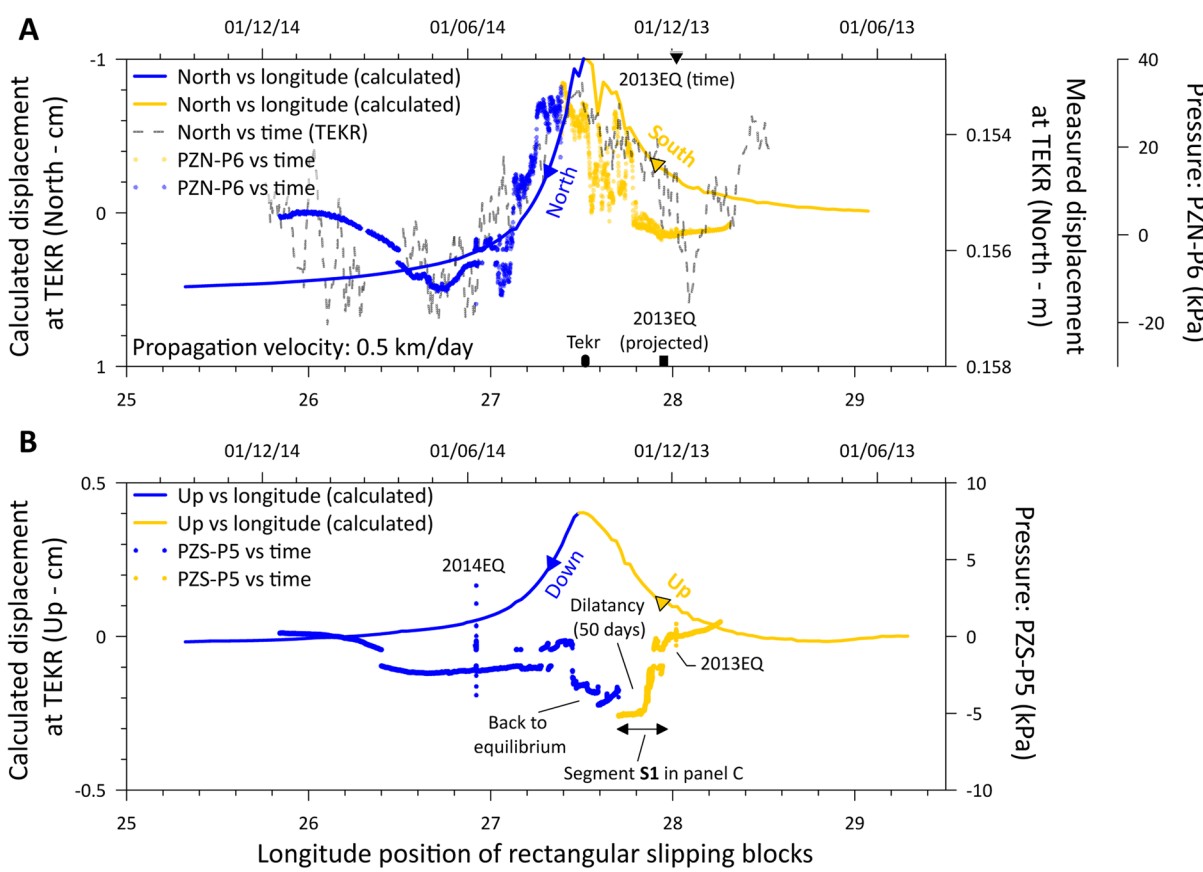

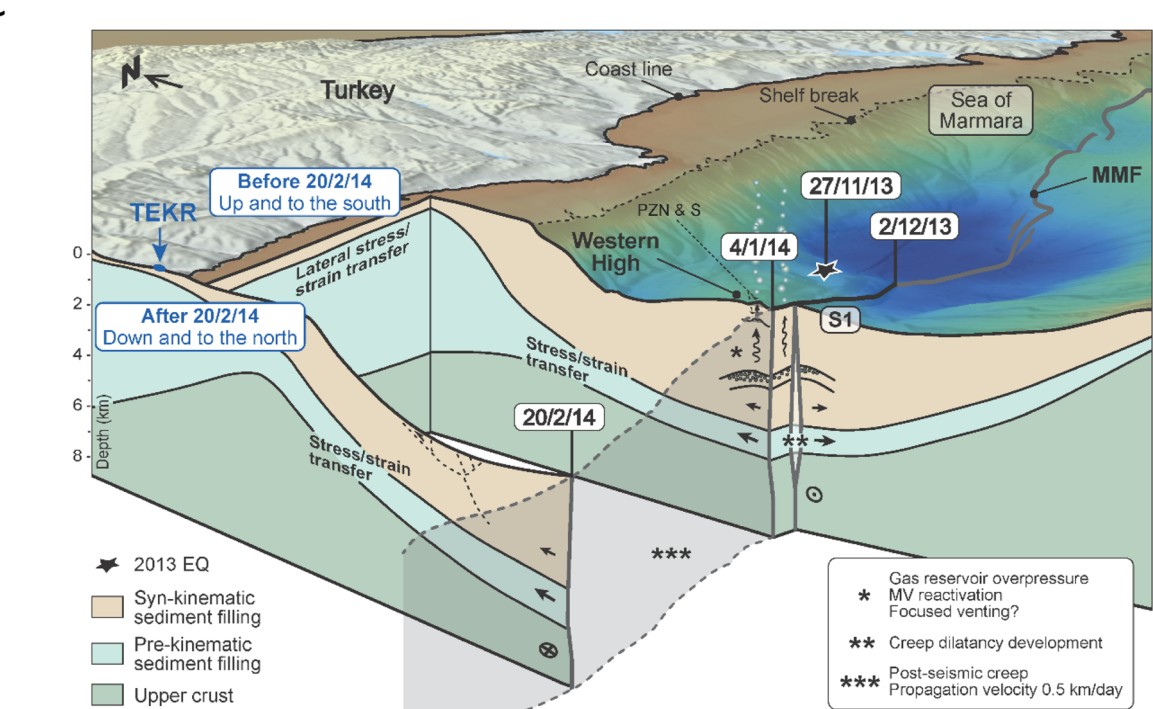

role of such pore fluid-sediment interactions in controlling or accompanying the process of fault slipping.

## Discussion

Taken all those observed and modeling results together, it is possible to draw a timeline of events that occurred after the 2013EQ indicating the way the aseismic creep affects the MV activities as well as the 3D displacement field surrounding the MMF (Fig. 5C). Shallow slipping along the MMF (within the syn- and pre-kinematic layers[42]) propagating from east to west is probably at the origin of the far-field observed geodetic data from TEKR station: up and south displacements before 20/02/2014 and down and north displacements after this date (Fig. 5C). The

**Fig. 5 Time line scenario. A** Calculated north displacement at TEKR versus longitude position of slipping compared to observed north displacement versus time. Yellow curves correspond to the displacement towards the south while the blue curves correspond to the displacement toward the north. Pore-pressure data from PZN-P6 versus time are also shown with an increase in the pore pressure during the first stage corresponding to the increase of the normal stress at the outer limit of the damaged zone (yellow dots). Earthquake (2013EQ) timing is indicated by a triangle (top horizontal axis) and its position is marked by a black rectangle (base horizontal axis). The TEKR geodetic station is indicated by a black rectangle (base horizontal axis). **B** Calculated up displacement at TEKR (yellow corresponds to up and blue to down) versus longitude position of slipping compared to pore pressure from piezometer PZS-P5 with a decrease of the pore pressure during the slip dilatancy affecting the piezometer area (yellow dots). A 50-day dilatancy period corresponds to a creep dilatancy along the segment S1. **C** 3D diagram illustrating the consequence of aseismic creep of the strike-slip fault on the mud volcano activities and the 3D displacement field surrounding the Main Marmara Fault (MMF). A black star indicates the 2013EQ earthquake epicenter and a blue dot the TEKR geodetic station. Locations of piezometers PZN and PZS are also indicated. The aseismic slipping propagates along the MMF at 0.5 km/day from east to west lasting >10 months.

shear-dilatancy caused by the slow slipping along the WH fault segment recorded by PZN-P5 between 2/12/2013 and 20/01/2014 corresponds to the reactivation of the MV by increasing the normal stress at the outer limit of the damaged zone. The MV pressurized by the increase of the normal stress is at the origin of the high pore pressure recorded by PZN-P6 (red dots in Fig. 5A). The subsequent decay of the pore pressure recorded by PZN-P6 (blue dots in Fig. 5A) reaching negative values is most likely the result of the decrease of the normal stress at the MV boundary causing the swelling of the MV and requiring the replacement of the dissipated pore-fluid volume.

This aseismic slipping event propagates along the MMF at 0.5 km/day from east to west lasting more than 10 months. However, based solely on the available data, it is not possible to ascertain when or where this 10-month SSE has nucleated. It may have started with the 2013EQ or alternatively the 2013 earthquake may have been triggered by the SSE. The slipping event we observe occurs mainly along the WH segment, a region having the largest proportion of mainshocks with associated foreshocks and aftershocks[43]. According to Martinez-Garzon and co-authors[43], the WH segment is close to failure increasing therefore the susceptibility of the segment to seismic triggering. Our results confirm also what was observed by Caniven and co-authors[44], indicating based on numerical calculations that fault dilation is a "*key factor controlling fault slip stability*."

Comparable features have been observed for SSE on subduction zones[45]. Shorter duration (3–20 days) SSEs have been detected along the San Andreas fault, however only at depth (>16 km)[46]. It has been proposed that SSEs have been linked with the migration of pore-pressure waves in subduction zones[47] and strike-slip faults[48] caused by fault valving. The transient nature of the observed inflation at PZN and PZS as well as the geodetic displacement obtained from a single GPS station (TEKR) provide direct evidence supporting this proposition.

Beyond the present case concerning the MMF, our data provide valuable information on the kinematics of transform faults. The deployment of surface piezometers measuring pore pressure all along a transform fault can help define the time scale regulating the coupling between faulting, creeping, and dilatancy processes. MVs are common features affecting the seafloor, particularly at subduction zones, where the relationship between MV plumbing systems and slow-slip mechanisms is an emerging field in earthquake research. MV instrumentation with surface piezometers may provide valuable information on understanding the shear-strain coupling at depth, within the seismogenic zone. The pressure drop recorded 4 days before the 2013EQ suggests that such an experiment could be a source of improving scientific knowledge in earthquake nucleation processes. We also show that new insights can be gained from the detailed observation of onshore geodetic data since small disturbances seem to have major meaning in terms of transient deformations along active faults.

## Methods

**Seabed amplitude and sub-seabed seismic features**. Site surveys including high-resolution 3D seismic and AUV bathymetric data were used[15] to draw the seabed amplitude and sub-seabed seismic features characterizing the study area (Supplementary Fig. 1).

**Geodetic data**. The 7-year period GPS horizontal displacement (east with respect to fixed EU plate) time series[6] shows that the Anatolian plate evolves smoothly to the west relatively independently of 2013EQ (Supplementary Fig. 2). Horizontal displacements (east) recorded by the two stations from the Eurasian plate (ISTN and TEKR) remain almost unchanged during the monitoring period. The 2014EQ unequally affects data from BALK, YENC, and CANA stations.

**Tools: piezometer and coring**. Free-fall piezometer equipped with a sediment-piercing lance of 60 mm diameter[49] is used to perform in situ pore pressure and temperature measurements. The piezometer is ballasted with lead weights (up to 1000 kg) and equipped with differential pore pressure and temperature sensors. Two piezometers were deployed within the shallow sub-surface (<7.08 mbsf) at sites PZN and PZS. The piezometer pore pressure and temperature sensors have an accuracy of ±0.5 kPa and 0.05 °C, respectively (data in Supplementary Figs. 3–8) Core MNT-KS26 was recovered at 300 m to the west of PZS (Fig. 1). The total length of the core is around 8 m. Density and P-wave velocity data were obtained on undisturbed sediment sections using the GEOTEK core logging devices (MSCL). Supplementary Fig. 9a, b shows the density and P-wave velocity versus depth. Intervals characterized by high density and/or high P-wave velocity anomalies are indicated as gray areas and could correspond to coarser materials. The effective lithostatic stress profile shown in Supplementary Fig. 9c was calculated from the density profile.

**Pore-pressure and temperature data**. Pore-pressure and temperature data from PZN recorded for more than 1 year are shown in Supplementary Fig. 3. The pore pressure generated by the piezometer penetration decays smoothly during several weeks and stabilizes for the shallowest four sensors (Supplementary Fig. 3a). Pore-pressure data from the two deepest two sensors fluctuate throughout the monitoring period (Supplementary Fig. 3a). Negative pore pressures as low as −13 kPa were recorded by the deepest sensor (Supplementary Fig. 3a). The temperature decays rapidly to reach in situ equilibrium temperature (Supplementary Fig. 3b). The temperatures at the deepest three sensors intersect regularly and are almost constant throughout the monitoring period.

Supplementary Fig. 4 shows an enlargement of pore-pressure generation during 2013EQ (Supplementary Fig. 4a) and 2014EQ (Supplementary Fig. 4b). During 2013EQ, pore-pressure data from the 6 sensors were affected for a few hours and fluctuates between −4 and +4 kPa. Despite the large distance of PZN from the 2014EQ epicenter (Fig. 1), the earthquake significantly affects the pore pressure from the 6 sensors which fluctuate between −25 and +8 kPa (Supplementary Fig. 4b).

Pore-pressure and temperature data recorded for >1 year at PZS are shown in Supplementary Fig. 5. The pore pressure and temperature generated by the piezometer penetration decays rapidly after installation (Supplementary Fig. 5a). Pore-pressure data from P5 fluctuate between −8 and 0 kPa throughout the monitoring period (Supplementary Fig. 5a).

Supplementary Fig. 6 shows an enlargement of pore-pressure generation during 2013EQ (Supplementary Fig. 6a) and 2014EQ (Supplementary Fig. 6b). Pore pressure data from the 6 sensors were affected by both events and fluctuates between −1 and +1 kPa during 2013EQ (Supplementary Fig. 6a) and between −7 and +8 kPa during 2014EQ (Supplementary Fig. 6b).

Thermal and hydraulic gradients plotted in Supplementary Fig. 7 correspond to ten different periods of monitoring. Thermal gradients at PZN (Supplementary Fig. 7a) suggest that the temperature field at the level of the upper four sensors is in a permanent regime and primarily diffusion-controlled (quasi-constant gradient) while it is advection-controlled along the deepest two sensors (quasi-constant temperature). Hydraulic gradients shown in Supplementary Fig. 7a indicate that

the pore pressure perturbation only concerns the two deepest sensors. Thermal data from PZS (Supplementary Fig. 7b) indicate a transient temperature regime without the possibility to conclude about the thermal process (advection or diffusion) controlling the temperature field. The non-linearity of the temperature profile could be the result of a transient diffusion of advection processes. The hydraulic gradients at PZS show that only the sensor P5 at 6.28 mbsf is concerned by the pore pressure perturbations.

The measured initial pore water pressure generated by the rod insertion ($\Delta u_i$) depends on the shear strength and the elastic properties of the medium[50]. Data in Supplementary Fig. 8a show a general linear increase of $\Delta u_i$ with depth except for PZS-P5 and PZN-P6 deviating slightly from this trend and indicating stiffer materials. The hydraulic diffusivity $C_h$ (or the horizontal coefficient of consolidation) of the medium normalized by the square root of the rigidity index ($\sqrt{I_r}$) was derived by using the cavity expansion theory[49] and by back-analyzing the pore-pressure decay curves following the piezometer penetration. Comparisons between the measured and the calculated pore pressure decay at PZN and PZS are shown, respectively, in Supplementary Fig. 8c, d.

**Transient diffusion–advection process**. To understand how pore-pressure variations at PZN-P6 and fluid advection may affect the pore-pressure field in the above sedimentary layers, a one-dimensional (1D) transient diffusion–advection equation (Eq. 1) is solved numerically using a finite difference method.

$$\frac{\partial u}{\partial t} = \frac{\partial}{\partial z}\left(C\frac{\partial u}{\partial z}\right) + v_z\frac{\partial u}{\partial z} \quad (1)$$

In Eq. (1), $C$ signifies the hydraulic diffusivity of the sediment, $v_z$ the vertical velocity of the fluid, $u$ the pressure field, $t$ the simulation time, and $z$ the depth below the seabed. To solve numerically the 1D diffusion–advection equation, a centered explicit finite difference discretization scheme is used. In this calculation, $C$ was taken equal to $1.5 \times 10^{-8}$ m$^2$/s equivalent to the $C_h$ values at PZN-P6 (Supplementary Fig. 8b) and for a rigidity index $I_r$ of 40. Three values of $v_z$ are considered showing that pure diffusion ($v_z = 0$ m/s) is needed to reproduce the general trend of PZN-P5 data while $v_z$ value of $3 \times 10^{-8}$ m/s is required to simulate the observed peaks at PZN-P5. For the considered $C$ and $v_z$ values, the PZN-P6 imposed pressure affected none of the levels above PZN-P5. Those results confirm that the hydraulic process controlling the pore-pressure field above PZN-P6 is an alternation between pure diffusion and diffusion–advection and this depends on the pressure level and hydro-fracturing at PZN-P6.

**Pore pressure and hydro-fracturing**. Hydro-fracturing in sediments takes place when the excess pore pressure exceeds the least principle stress plus the tensile strength of the medium. Comparing recorded pore pressure at PZN-P5 (Supplementary Fig. 11a) and PZN-P6 (Supplementary Fig. 11b) with effective lithostatic stresses ($\sigma'_v$) at corresponding depth indicates that PZN-P6 pore-pressure peaks are reaching 70–100% of $\sigma'_v$ before dropping sharply. These high pore-pressure values suggest that hydro-fracturing occurred at PZN-P6 level during the first 5 months of monitoring. In contrast, the pore pressure at PZN-P5 is too low to initiate hydro-fracturing.

**3D displacement field in the half-space linear elastic medium due to shear and tensile along the MMF**. The aim of this modeling was to ascertain whether the anomalous deformation observed at the geodetic station TEKR could be explained by a slipping event on the MMF and if so, provide some constraint on it. To do this, a 3D elastic half-space dislocation model[37] with a Poisson ratio of 0.25 was used to model the surface displacement. This model is based on elastic dislocation theory whereby the displacement in the $i$ direction at position $x$ due to a dislocation $d$ on a planar fault subdivided into $N$ individual cells $\Gamma(l)$:

$$\Delta u_i(x) = \sum_{l=1}^{N}\left(\int_{\Gamma(l)} T_{ik}^{ss}(x,\xi_l)\,n_k d_{ss}(\xi_l)dS + \int_{\Gamma(l)} T_{ik}^{t}(x,\xi_l)\,n_k d_t(\xi_l)dS\right) \quad (2)$$

where $n_k$ is the normal vector to the fault plane and $d_{ss}$ and $d_t$ are shear and tensile slip in cell $i$, respectively. $T_{ik}^{ss}(x,\xi_l)$ and $T_{ik}^{t}(x,\xi_l)$ map the shear and tensile slip at position $\xi$ to the displacement in the $i$ direction at position $x$[51] using the method of images with analytical description of deformation in an elastic medium, these kernels have explicit formulas for both shear and tensile rectangular sources[52]. The principle of superposition allows different source mechanisms and slipping surfaces to be calculated individually and summed together to produce the total displacement. As a result, the digitized fault trace[11] of the MMF is subdivided into segments no larger than 2 km. Assuming the MMF has a dip of 90°, a discretized fault plane was constructed with individual slipping cells no larger than 2 km × 2 km producing 158 × 7 non-overlapping cells. Two faulting mechanisms are considered (a) permanent pure strike slip (i.e., $d_t = 0$) and (b) a permanent strike slip with a transitory tensile component in the active cell (i.e., $d_t$ is a box-car function in time while $d_{ss}$ is a Heaviside function). The choice of pure-right lateral slip is in keeping with the regional geodetic motion[31] as well as the historical seismicity[14]. The inflation contributions from the different elements are not summated as we assume, based on observations at PZN and PZS, that inflation is a transient phenomenon only active during slipping.

To investigate the propagation of a slipping front across the fault, the contribution from the cells is summed sequentially based on geographical location (i.e., from east to west and from west to east) allowing for the assessment that different sections of the fault make the displacement at TEKR. Therefore to ascertain the displacement in the direction $i$ at TEKR due to slipping event propagating from east to west where the active slipping cell is cell $n$, Eq. 2 can be rewritten as:

$$u_i^{EW}(n) = r\sum_{i=1}^{n\leq N} \Delta u_i^s + (r-1)\Delta u_i^t \quad (3)$$

where $\Delta u_i^s$ is the displacement due to 1 m strike slip of cell $i$ (i.e., the first integral in Eq. (2)), and $\Delta u_i^t$ is the contribution from the 1 m of tensile displacement of cell $i$ with the cells ordered sequentially from east to west. $r$ represents the ratio of strike to tensile slip. Using the same formalism, west to east deformation is defined as:

$$u_i^{WE}(n) = r\sum_{i=1}^{n\leq N} \Delta u_{N+1-i}^s + (r-1)\Delta u_{N+1-i}^t \quad (4)$$

Sensitivity to depth was tested by propagating the slipping event across the fault at a constant depth for each simulation with a width of 1 cell (i.e., 2 km). The procedure for west-to-east simulation is depicted in Supplementary Fig. 12. In summary, four cases are considered, Case: A; a purely right-lateral rupture propagates from east to west; Case B; a purely right-lateral rupture propagates from west to east; Case C: a right-lateral rupture with a tensile component propagates from east to west; Case D: the right-lateral rupture with a tensile component propagates from west to east. Within each of these cases, the slip surface has been placed at a range of different depths.

With only one observation point, we have focused our analysis on comparing the simulated general trends of the vertical and north–south displacement at the location of TEKR with the observations. The key features we are looking to observe are an initial displacement to the north followed by a motion to the south, and on the vertical component, an upward displacement followed by subsidence. Supplementary Table 1 summarizes all the simulations and whether they conform to the aforementioned trends while Supplementary Figs. 13–16 depict all simulated ground motion at TEKR. As shown in Supplementary Table 1, the only simulations that match these trends is the case where the rupture propagates from east to west and rupture occurs at a shallow depth (i.e., 0–6 km in the pure strike slip case, and 0–8 km in the mixed strike slip–tensile rupture).

In order to produce the observed southern displacement of 0.35 cm at TEKR, this would require a right-lateral and dilatant displacement of 0.6 cm; and 0.2 cm in the case of a pure strike-slip movement, which are equivalent to M5.3 and M5.1 earthquakes, respectively (using a shear modulus of 30 GPa and considering them as one large continuous event). However, changing the element size (e.g., increasing/decreasing cell size, including multiple cells from different depths with different slips at the same time) or the inclusion of normal faults in the vicinity of MMF may change these values. Consequently, no conclusion is drawn on the size of the active slipping zone.

The distance between the barycenter of the sources is converted to time by assuming a constant rupture velocity for the slipping event, the value that provided the best fit to the observations was a velocity of 0.5 km/day.

## Data availability
GPS data are freely available at Nevada Geodetic Laboratory (http://geodesy.unr.edu) and piezometer data can be downloaded from https://doi.org/10.17882/79781. Geological (AUV bathymetry, seismic lines) and sedimentological data are available in the main text or the Supplementary Materials.

## Code availability
Computer code used to generate results that are reported in Supplementary Figs. 8, 10, and 13–16 are available at https://github.com/nsultan-2021/advection-diffusion. Surface displacement calculations can be found here: https://github.com/s-murfy/StrikeSlipDef

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

## Acknowledgements

The authors thank Nurcan Meral Özel (KOERI), P. I. of MARSITE project. Luca Gasperini (CNR), led the deployment cruise (MARMARA-2013) of R/V Urania. Livio Ruffine, Namik çagatay, and Pierre Henry co-led the recovery cruise (MARSITECRUISE) of R/V Pourquoi pas?. Bruno Marsset, Yannick Thomas, and Stephan Ker are thanked for providing the 3D seismic data. The marine operations for instrument deployment, recovery, and analyses were supported by the 7th EU-funded Framework Programme (FP7), within EMSO and MARSITE (EU Grant agreement ID: 308417) Projects and ANR MAREGAMI project (Grant No. ANR-16-CE03-0010). Supplementary Fig. 12 is generated with PyGMT[53].

## Author contributions

N.S. led the piezometer and geodetic data analyses and wrote the manuscript. S.M. carried out the 3D numerical calculations to derive the 3D displacement field at TEKR and contributed to data analyses and interpretations. V.R. carried out the geological analyses and contributed to data interpretations. L.G. conceived and designed the off-shore experiment and contributed to data analyses and interpretations. S.M., V.R., and L.G. helped write the manuscript.

## Competing interests

The authors declare no competing interests.
