## [Peer Review File · Nature Communications]

REVIEWER COMMENTS

Reviewer #1 (Remarks to the Author):

Review to: Creep-dilatancy development at a transform plate boundary

By Nabil Sultan, Shane Murphy, Vincent Riboulot, Louis Géli

In this manuscript, the authors investigated the mechanism generating a slow slip event in the Main Marmara Fault segment of the North Anatolian Fault, Turkey. They employed GPS time series to resolve a potential slow slip event lasting for about 10 months. In addition, authors deployed two submarine piezo sensors close to the fault and submarine mud volcano. They utilized the temporal evolution of pore fluid pressure data from them and identified that in one of them, pore pressures dropped below hydrostatic conditions coinciding with the occurrence of the 10-slow slip. The authors utilized this observation of pore pressure decrease to infer that the mechanism linked with the nucleation of the slow slip event could be creep-dilatancy. This mechanism has been previously suggested from numerical modelling studies and experimental, but not clearly verified.

The article is well written and constructed, with relatively minor typo errors. Unraveling the mechanisms behind the nucleation of slow slip events vs typical seismic events is unquestionably an important topic that deserves research. However, I have strong concerns about the scientific credibility of the results here presented. Specifically, neither the piezo sensors measuring pore fluid pressure signals nor the GPS are sufficiently processed to ensure that the observed signals go beyond seasonal fluctuations. Since the piezo sensors will be recording the absolute pressure, the pore pressure signals will contain a lot of unwanted information: barometric pressure, Earth tides, Ocean tides, tsunamis, surface waves etc. Maybe the authors did all this processing but this is not at all described in the article. Therefore, it seems plausible that the signals that are here correlated are just seasonal fluctuations. Additional comments are provided below.

GPS data: Showing seasonal variations in the GPS data is clearly not sufficient. I agree that it seems that at TERK station, after the 2013 earthquake, there may be some signal superimposed with the signal, but this must be recovered much more accurately. Seasonal variation of the GPS are good to be shown in the Supplementary materials, to demonstrate that the stations are recording coherent signals, but in the main analysis the GPS data needs to be clean from seasonal variations and showing clearly the slow transient and its dimension, for all GPS stations.

Piezo data: What is the depth of the sea at this location, or, in other words, the height of the water column on top of the sensors? No seasonal variations are reported here to affect the pore pressure measurements. This is surprising, as pore fluid pressure sensors at the sea bed will likely correlate with the height of the water column of top of the sensor, which will strongly vary with e.g. tidal signals. To proof the correct functionality of the PZ data, it is encouraged that records of seasonal and/or tidal variations and their amplitude on the piezo recordings are added to the supplementary materials and compared with the here reported signal after the 2013 earthquake.

Additional implications/verifications from the creep dilatancy mechanism: this proposed mechanism for the nucleation of the slow slip even has some implications for the permeability and diffusivities of the Marmara Fault. Could you establish, based on your observations, a range of permeability values that would be compatible with it, and whether they are physically plausible or not?

Specific Comments according to line number:

L#22: How does the pore pressure at the shallow sea bed relates to the pore pressure of the fault at depth? This is not obvious, even with the presence of a mud volcano nearby.

L#24: Because of the poor processing of GPS data, the 10-month slow slip is yet to be properly identified.

L#44,45: Repeaters have also been characterized in this region by Bohnhoff et al., (Geophysical Journal International 2017) and Yamamoto et al (Tectonophysics, 2020).

L#62 onwards: There is no explanation provided about the processing applied to the data from piezo sensors. Which processing is done to the data shown in Fig 2b? As the pore pressure sensors are deployed on the sea bed, I expect that they are effectively measuring the height of the water column above. As the Sea level varies according to seasonal and tidal effects, I suggest to include in the supplementary materials some figures illustrating the correct recording of these signals and their amplitude. This way, the rest of the observed signals would gain credibility and we could compare the corresponding amplitudes.

L# 64-66: Please specify the epicentral and hypocentral distances from the 2013 and 2014 earthquakes to each of the PZ sensors.

L# 70, Fig 2a: The GPS data needs to be corrected by seasonal variations, and then we will see what is left and how many stations show the transient and what is the displacement.

L#72,74: Similar triggered slow slip events have been observed in the eastern Marmara region (Martinez-Garzon et al, Earth and Planetary Science Letters 2019). Are there similarities with these observations?

L#86. This period of observed higher pore pressure at PZN should result in the creation of new fractures. Is the seismicity data supporting this?

Also, after the initial three months, the pore pressure decreased at PZN, going below hydrostatic conditions. This seems quite similar to what was recorded at PZS during the slow slip event and here interpreted in the frame of creep dilatancy. Why is it here not interpreted in the same way?

L#101-102: This needs to be much better shown once the seasonal signal is out.

L#103-105: The “strong link” here mentioned could likely just be due to the seasonal signals dominating both types of data records, and here not mentioned for the piezo sensors PZN and PZS.

L#116-118, I dare to disagree with this statement as well. GPS data as such only reflects displacement at the surface. It is only by modelling and inferring a locking depth that transients over a depth range can be effectively recovered. As the pore pressure sensors are such few meters below sea bed, I really don't see how the current data relates to the processes at depth.

L#119-121, Fig 2c, if the 2013 earthquake nucleated closer to the PZS, why is it only the PZN sensor close to the mud volcano the one that shows a signal before the nucleation of the earthquake?

Fig 3b: I think the correlation shown in this plot could also be due to the fact that both types of data are dominated by seasonal signals. Could you more firmly exclude this possibility ?

L#136: Is Fig S9 the right reference here? I cannot see the mentioned pore pressure evolution. Also, why would the P5 only be indicative of a generalized process in the region, if such trend cannot be seen in any of the other sensors from PZS or PZN? Is the data from all the other sensors coherent in between? How can we know that P5 is not recording some unrelated disturbance?

Section “Negative pore pressure and dilatancy”. A period with negative pore pressure is also seen at PZN. Why is it here not mentioned?

L#143-146: More parameters of the simulation are needed: what is the magnitude estimated of the slow slip event to match the observation? What are the elastic and friction parameters? The depth is only constrained to be < 8km. It would be optimal if the depth of the slow slip event could be better constrained.

L#167, not only for subduction zones, but even in the Marmara region, see comment from L#44.

L#170-171: But the inflation (dilatancy) is not seen in PZN, but only in one of the sensors from PZS, correct?

L# 180-182: The observed pressure drop 4 days before the 2013 earthquake is less than 1 kPa, therefore, it should lay in the range of the daily-fortnightly stress changes from tidal variations. Can such change just reflect a tidal variation in the sea level?

Reviewer #2 (Remarks to the Author):

Dear Editor,

hereby I send you my review of Sultan et al. paper entitled “Creep-dilatancy development at a transform plate boundary”.

The paper addresses a major scientific question, related to the mechanism(s) that allow plate boundary segments to accommodate a continuum of rupture speeds ranging from those of standard earthquakes (2-3 km/s) to relative plate motion (a few cm/yr). The authors propose/support the major role played by slip dilatancy in regulating tectonic slip.

The manuscript presents a direct evidence of pore pressure changes associated with an aseismic tectonic transient (i.e., slow-slip event) along a branch of the North Anatolian Fault in the Sea of Marmara (Main Marmara Fault). The interpretation is based on the observation of the coupling between pore pressure changes, detected by near-fault piezometers, and tectonic deformation, recorded at a GPS station.

I really enjoyed reading this paper, which is well written and organized, and the main results and concepts are nicely illustrated in the figures. Although the authors put together a very nice story, my only major concern is about the detection of the aseismic slip signal at a single station as I will discuss below in my comments.

Hereafter follow a main comment and several minor comments that could help to enhance the clarity of the study.

Main comment:

As already mentioned, my main concern is about the tectonic transient signal (i.e., slow slip event) recorded from a single station. Records from single stations leave always open the possibility that the signal is due to a very local disturbance and make very difficult, and often not unambiguous, their interpretation. To overcome this limitation the authors, perform numerical modelling to better understand the origin/characteristics of the signal recorded at the GPS station TEKR. The modelling results indicate that the detected signal is compatible with an aseismic slip episode propagating at 0.5 km/day from E to W at shallow depths (0-8 km), along the Main Marmara Fault.

The possible slow slip event is recorded on the N-component while it produces no detectable deformation on the E- component (GPS station TEKR). I would imagine a slow-slip event originated along an E-W striking transform fault to produce a larger signal on the E-component than on the N-component as also the results from the modelling seem to show (Fig. 13-16S). I think the authors

should discuss and explain this part a bit better (i.e., why the signal is visible on the N-component and not on the E-component?).

To my knowledge the vertical component of the GPS is the one affected by the largest errors and therefore the trickier to use. Is the +1 cm displacement deficit above the noise level of the data? Which is the noise level of the data? Did the authors correct the data for precipitation records in the region or sea level variations? (e.g., can it be excluded that the signal is generated by a local peak in the rainfall or by sea level oscillations?). In general, I believe that a more detailed explanation of the processing of the geodetic data is needed to give the reader a better idea about the corrections applied (e.g., seasonal trend, steps from earthquakes, precipitations) to the data and if the noise level of the data are smaller than the amplitude of the detected signal(s).

To strengthen the hypothesis of the tectonic origin of the geodetic signal at TEKR the authors could consider the following suggestions:

- The repeater families reported in Schmittbuhl et al. (2016) cover the temporal interval during which the authors detect the tectonic transient. Is there an acceleration of the relative plate motion indicated by the repeaters? If yes, this could be a strong argument in support of the tectonic origin of the signal at the location suggested by the authors. If not, then the authors should try to explain why the acceleration of the relative plate motion is not evident in the slip rates inferred by the repeater sequences.

- How the recorded signal in this study compares with other aseismic slip signals detected along the North Anatolian Fault in terms of propagation velocities and depth intervals? (e.g., Aslan et al., 2019; Rousset et al., 2016).

Minor comments:

- Fig. 1A: do the authors mean Fig. 2B (above the black arrow indicating the study region) or should it be Fig. 1B?

- Fig. 1B: coordinates are missing (same as Fig. 1A-C of the supplement).

- Fig. 4: It could be useful to also indicate the location of the piezometers and indicate the temporal variation of the pore pressure together with the deformation observed at station TEKR (already in the figure). Basically, to synthesize what the authors mention in ln. 111-113 the authors could use a

different color to show/indicate: (1) TEKR displacement towards South and pore pressure increase at PZN-P6, and (2) TEKR displacement towards North and pore pressure decrease.

- Concerning the effect of pore pressure changes and fault valve behavior in regulating tectonic slip I think that there are two recent papers that are omitted in the references both in the introduction (ln. 40) and in the discussion (ln. 170). Gosselin et al. (2020) and Warren-Smith et al. (2019) provide seismological evidence for fault-valve behavior proposing it as the mechanisms controlling the genesis of slow-slip earthquakes. Probably they should be integrated in the manuscript.

- I came across the paper of Proctor et al. (2020) about “direct evidence for fluid pressure, dilatancy, and compaction affecting slip in isolated faults”. How pore-pressure changes during slip dilatancy compare with the one presented in the Proctor et al. paper? This could be an interesting part to include in the discussions. In fact, the observations of the authors rely on a single occurrence which does not imply a repetitive occurrence of such behaviour, e.g., should we always expect a pore-pressure drop before the nucleation of an earthquake (ln. 119-121)? Laboratory experiments by reproducing multiple deformation cycles (e.g., Proctor et al., 2020) could help to address it.

- station TEKR is written both in capital and lowercase (e.g., ln. 147, ln. 159, Fig. 3-4), the authors may want to uniform it.

- ln. 114-115: “wet period” and “dry period” refer to the system (i.e., wet/dry conditions) or to the weather and therefore to the amount of precipitations? Needs to be clarified.

- ln 158: “incontestably” is the most appropriate term to use in this case?

- ln. 167: since there is evidence for creep bursts along the North Anatolian Fault it could make sense to refer also to such papers (Aslan et al., 2019; Rousset et al., 2016)

References:

Aslan, G., Lasserre, C., Cakir, Z., Ergintav, S., Özarpaci, S., Dogan, U., et al. (2019). Shallow creep along the 1999 Izmit earthquake rupture (Turkey) from GPS and high temporal resolution interferometric synthetic aperture radar data (2011–2017). *Journal of Geophysical Research: Solid Earth*, 124, 2218–2236. <https://doi.org/10.1029/2018JB017022>

Gosselin, J. M., Audet, P., Estève, C., McLellan, M., Mosher, S. G., & Schaeffer, A. J. (2020). Seismic evidence for megathrust fault-valve behavior during episodic tremor and slip. *Science advances*, 6(4), eaay5174.

Proctor, B., Lockner, D. A., Kilgore, B. D., Mitchell, T. M., & Beeler, N. M. (2020). Direct evidence for fluid pressure, dilatancy, and compaction affecting slip in isolated faults. *Geophysical Research Letters*, 47, e2019GL086767. <https://doi.org/10.1029/2019GL086767>

Rousset, B., Jolivet, R., Simons, M., Lasserre, C., Riel, B., Milillo, P., Çakir, Z., and Renard, F. (2016). An aseismic slip transient on the North Anatolian Fault, *Geophys. Res. Lett.*, 43, 3254–3262, doi:10.1002/2016GL068250.

Schmittbuhl, J., Karabulut, H., Lengliné, O., and Bouchon, M. (2016), Long-lasting seismic repeaters in the Central Basin of the Main Marmara Fault, *Geophys. Res. Lett.*, 43, 9527–9534, doi:10.1002/2016GL070505.

Warren-Smith, E., Fry, B., Wallace, L., Chon, E., Henrys, S., Sheehan, A., ... & Lebedev, S. (2019). Episodic stress and fluid pressure cycling in subducting oceanic crust during slow slip. *Nature Geoscience*, 12(6), 475-481.

Reviewer #3 (Remarks to the Author):

I find the manuscript very interesting and of critical importance in shedding light on the physics of important hydraulic phenomena relating deep processes to near-surface and surface observables. However I find the presentation of the material somewhat confusing as I indicated in my doc file.

The manuscript by Sultan et al. is a detailed analysis of tectonics-driven hydraulic phenomena in the Sea of Marmara to understand the mechanical interplay between the pore pressure variations and both earthquakes and deeper level strain transients. To this end they use records taken from the seabed in the Sea of Marmara and onshore geodetic measurements. The work is mainly targeting to understand the mechanisms through which near-surface observations of pore pressure and geodetic measurements are affected by the processes at depth. The relationship between the slow slip events and hydraulic phenomena have indeed been documented before (mostly for subduction zones), so in this context studying a well-documented faults zone such as the MMF is indeed very interesting. The authors detect a pore pressure transient preceding a seismic event and relate the polarity of the transient to the onland geodetic observations to conclude a, so to speak, teleconnection between both observables and the physical event through shear dilatancy which is basically the volume change observed in granular materials when they are subjected to shear deformations. In that respect, if shear dilatancy is indeed proven to be playing the role that the authors claim, then monitoring pore pressure transients continuously would indeed be very valuable to better understand seismicity.

The manuscript features two numerical models to account for the pore pressure and GPS data as mechanical response to both earthquakes and the preceding creep events.

I find the results very interesting, however I find the presentation of the manuscript confusing and not properly sequenced, forcing the reader to concentrate on several aspects of the phenomena simultaneously. The effect of the deep tectonic processes in causing the surface elastic vertical displacements is discussed without a proper explanation of the geometry. A schematic figure to

explain the geometry of the numerical model (together with a proper explanation of the boundary conditions) is necessary. I would also be willing to see the numerical code used for both the elastic displacement and advection-diffusion. Also missing is a supplement of the mathematics that is used for this model with at least some detail in the numerical method. The formulation of the pressure transients in terms of the advection-diffusion equation is better discussed but the link between this model and the surface elastic phenomenon is “lost in translation”. The slip scenarios discussed at the end of the text indeed shed light on the possible effect on the geodetically measurable observations but a scale analysis is missing to relate the results to quantitatively tie to the pressure modeling (and observations). A better structured text would definitely help this aspect. I find the approach logical but difficult to follow. Another confusing aspect is the fact that the temperature data is discussed in a very qualitative way. The way in which the temperature field is coupled with the ongoing pressure transients must be discussed using the basic thermodynamics of the granular system under consideration. Furthermore some statements were given without a proper explanation, especially in the discussion of demarcating the zones dominated either by diffusion or advection. For instance in the statement below:

“Thermal gradients at PZS (Supplementary Fig. 7b) do not allow concluding about the thermal process controlling the temperature field while the hydraulic gradients at PZS show that only the sensor P5 at 6.28 mbsf is concerned by the pore pressure perturbations “

Here it is not clear what is meant by “thermal process”. It is also not clear why the gradients at the PZS do not allow whether or not these thermal processes control the temperature field. I am not asking the author to use a n additional energy equation to couple their existing numerical model to properly model the temperature field but I expect clearer explanations.

All in all this is a very interesting manuscript but a reorganization of the text with the issues mentioned above is necessary.

The authors thank the referees for their very constructive comments. In the following, we give a point-to-point reply (blue) to the referee comments.

The line number when it is mentioned refers to the NCOMMS-21-23282-T-A_annotated.pdf.

Reviewer #1	
Comments	Reply
1. The article is well written and constructed, with relatively minor typo errors. Unraveling the mechanisms behind the nucleation of slow slip events vs typical seismic events is unquestionably an important topic that deserves research.	We thank the reviewer for this positive comment concerning the importance of the subject treated by the manuscript.
2. However, I have strong concerns about the scientific credibility of the results here presented. Specifically, neither the piezo sensors measuring pore fluid pressure signals nor the GPS are sufficiently processed to ensure that the observed signals go beyond seasonal fluctuations.	For the pore pressure measurements and as it is clearly indicated in the manuscript (line 82 in the old manuscript version), we are measuring differential pore pressure. Therefore, no correction is needed to consider the seasonal fluctuations. Data from figure 2B (PZS-P5) shows clearly the flatness of the signal after the 2014EQ during several months and independently of seasonal fluctuations. A sentence is added in line 66 to make it clear that the piezometer measures a differential pressure. We agree with the second comment concerning the GPS data and the shown data in the new version are now corrected for seasonal fluctuations (see lines 76 to 92 and figures 2 to 4). The interpretations remain valid with the new processed GPS data.
3. Since the piezo sensors will be recording the absolute pressure, the pore pressure signals will contain a lot of unwanted information: barometric pressure, Earth tides, Ocean tides, tsunamis, surface waves etc. Maybe the authors did all this processing but this is not at all described in the article. Therefore, it seems plausible that the signals that are here correlated are just seasonal fluctuations. Additional comments are provided below.	In fact, the piezometer is measuring differential pore pressure so no correction is needed (see above - #2).
4. GPS data: Showing seasonal variations in the GPS data is clearly no sufficient. I agree that it seems that at TERK station, after the 2013 earthquake, there may be some signal superimposed with the signal, but this must be recovered much more accurately.	In the new version of the MS, we used the STL (seasonal-trend decomposition based on LOESS) procedure in the software R. The STL is a filtering procedure, allowing to decompose a time series into three components: trend, seasonal, and remainder parts (see lines 76 to 92 and figures 2 to 4).

Seasonal variation of the GPS are good to be shown in the Supplementary materials, to demonstrate that the stations are recording coherent signals, but in the main analysis the GPS data needs to be clean from seasonal variations and showing clearly the slow transient and its dimension, for all GPS stations.	
5. Piezo data: What is the depth of the sea at this location, or, in other words, the height of the water column on top of the sensors? No seasonal variations are reported here to affect the pore pressure measurements. This is surprising, as pore fluid pressure sensors at the sea bed will likely correlate with the height of the water column of top of the sensor, which will strongly vary with e.g. tidal signals. To proof the correct functionality of the PZ data, it is encouraged that records of seasonal and/or tidal variations and their amplitude on the piezo recordings are added to the supplementary materials and compared with the here reported signal after the 2013 earthquake.	The piezometer is measuring differential pore pressure. This is now clearly indicated in lines 66-67. Water depths are added to the text (line 69-70).
6. Additional implications/verifications from the creep dilatancy mechanism: this proposed mechanism for the nucleation of the slow slip even has some implications for the permeability and diffusivities of the Marmara Fault. Could you establish, based on your observations, a range of permeability values that would be compatible with it, and whether they are physically plausible or not?	Hydraulic diffusivities from both piezometers are already calculated and are shown in the Supplementary Fig. 8B. The determination of the evolution of the hydraulic diffusivities is unfortunately not possible with the present available data.
7. L#22: How does the pore pressure at the shallow sea bed relates to the pore pressure of the fault at depth? This is not obvious, even with the presence of a mud volcano nearby.?	This is one of the main conclusion of the paper. Because pore pressure data show a signal similar to the geodetic data we conclude that the source of both perturbation is stress/strain at the level of the fault. A small perturbation of the pore pressure in the shallow sediments (< 8 mbsf) cannot be detected by onshore geodetic data unless the source is much deeper. Therefore, at this stage, it is not possible to conclude about the relationship between the pore pressure at the fault level and the observed ones based on our shallow piezometers we can just hypothesis that measured pore pressures at PZN-P6 and

	onshore geodetic perturbations have the same source.
8. L#24: Because of the poor processing of GPS data, the 10-month slow slip is yet to be properly identified.	With the new GPS processed data, we can still confirm the 10-month slow slip event (see fig. 4b).
9. L#44,45: Repeaters have also been characterized in this region by Bohnhoff et al., (Geophysical Journal International 2017) and Yamamoto et al (Tectonophysics, 2020).	Bohnhoff (2017) reports deep-seated repeaters for the 2006-2010 recording period hence (no overlap with our recording period) below the Central Basin and WH, with inter-event time of 12 months and 38 months, at depth of 7.8 and 6 km, respectively. Additional repeater pairs of smaller magnitude may have been missed, as the catalogue magnitude of completeness M_c is 2.7. Yamamoto et al (2019) provide evidence of creep along the Western High segment, based on acoustic telemetry, amounting to nearly half of the Anatolian/Eurasian slip rate. They show that "a simple model of three elastic layers—a partially locked / partially creeping sedimentary layer (8 km) at the top with the observed rate, a fully locked (3 km) layer in the middle, and a fully creeping bottom layer— reasonably explains the GNSS data". However, the best fitting thickness of the slipping patches we model is shallow (< 8 km), consistent with Yamamoto et al (2019); Rousset et al (2016) and Aslan et al (2019) results. This is now indicated in the new version of the manuscript (lines 177 and 178).
10. L#62 onwards: There is no explanation provided about the processing applied to the data from piezo sensors. Which processing is done to the data shown in Fig 2b? As the pore pressure sensors are deployed on the sea bed, I expect that they are effectively measuring the height of the water column above. As the Sea level varies according to seasonal and tidal effects, I suggest to include in the supplementary materials some figures illustrating the correct recording of these signals and their amplitude. This way, the rest of the observed signals would gain credibility and we could compare the corresponding amplitudes.	No correction is needed since we are measuring differential pore pressure (see above - #2).
11. L# 64-66: Please specify the epicentral and ded hypocentral distances from the	Epicentral distances from PZN are now included in the paper: 12.5 km from the 9 km deep, 2013EQ (Wollin et al, 2017) and 209

2013 and 2014 earthquakes to each of the PZ sensors.	km from the 11 km deep, 2014EQ (Saltogianni et al, 2015). See lines 74 to 75.
12. L# 70, Fig 2a: The GPS data needs to be corrected by seasonal variations, and then we will see what is left and how many stations show the transient and what is the displacement.	The correction is done and the discussion considers now the corrected GPS data for the N-S component. For the up-down direction, the non-corrected data seem more illustrative concerning the deviation of the signal from the general up-down tendency. Our initial interpretations remain valid with the new processed GPS data.
13. L#72,74: Similar triggered slow slip events have been observed in the eastern Marmara region (Martinez-Garzon et al, Earth and Planetary Science Letters 2019). Are there similarities with these observations?	Martinez-Garzon et al (2019) find that the WH region -where earthquake repeaters are interpreted as indicator for fault creep- also has the largest proportion of mainshocks with associated foreshocks and aftershocks, potentially indicating that this segment is closer to failure and has increased susceptibility to seismic triggering. We added a short paragraph in the new manuscript about the Martinez-Garzon et al. observation concerning the WH segment (see lines 216 to 219).
14. L#86. This period of observed higher pore pressure at PZN should result in the creation of new fractures. Is the seismicity data supporting this? Also, after the initial three months, the pore pressure decreased at PZN, going below hydrostatic conditions. This seems quite similar to what was recorded at PZS during the slow slip event and here interpreted in the frame of creep dilatancy. Why is it here not interpreted in the same way?	The absence of a seismometer installed in the near vicinity of the piezometers does not allow us to identify the acoustic response of the propagation of such a fracture in a soft and superficial sediment. The decrease of the pore pressure at PZS-P5 is abrupt (see Figure 5b for instance) indicating a sudden external mechanism disturbing the pressure while the pore pressure at PZN-P6 decreases to negative values by following a gentler curve. The data from PZN-P6 fit well with the model we tested with mainly the following scenario: The MV pressurized by the increase of the normal stress during shear-dilatancy is at the origin of the high pore pressure recorded by PZN-P6 (red dots in Fig. 5a). The subsequent decay of the pore pressure recorded by PZN-P6 (blue dots in Fig. 5a) reaching even negative values is most likely the result of the decrease of the normal stress at the MV boundary causing the swelling of the MV and requiring replacing the dissipated pore-fluid volume.

	This is added to the discussion paragraph (see lines 205 to 209).
15. L#101-102: This needs to be much better shown once the seasonal signal is out.	Ok done, see # 12
16. L#103-105: The “strong link” here mentioned could likely just be due to the seasonal signals dominating both types of data records, and here not mentioned for the piezo sensors PZN and PZS.	The piezometer is measuring differential pore pressure, so no correction is needed.
17. L#116-118, I dare to disagree with this statement as well. GPS data as such only reflects displacement at the surface. It is only by modelling and inferring a locking depth that transients over a depth range can be effectively recovered. As the pore pressure sensors are such few meters below sea bed, I really don’t see how the current data relates to the processes at depth.	This is the main point of the paper. On one hand piezometers are measuring differential pore pressures within the shallow sediments (< 8mbsf) and on the other hand onshore geodetic data are measuring a more regional displacement field. The similarity between the two signals let us suppose that the source is the same and since it is affecting two different instruments at 35 km distance, we conclude that both are detecting a deep process. A shallow localized process at the level of the piezometer is impossible to be detected by onshore GPS data and vice versa.
18. L#119-121, Fig 2c, if the 2013 earthquake nucleated closer to the PZS, why is it only the PZN sensor close to the mud volcano the one that shows a signal before the nucleation of the earthquake?	Because we believe that, the MV acts as a window to the MMF seismogenic zone linking stress/strain changes at depth to shallow pore-pressure variations (lines 118-119 in the old version) and PZN-P6 is measuring pore pressure within this MV (see Figure 1C).
19. Fig 3b: I think the correlation shown in this plot could also be due to the fact that both types of data are dominated by seasonal signals. Could you more firmly exclude this possibility ?	Yes we can exclude definitely this hypothesis since the piezometer is measuring differential pore pressure.
20. L#136: Is Fig S9 the right reference here? I cannot see the mentioned pore pressure evolution. Also, why would the P5 only be indicative of a generalized process in the region, if such trend cannot be seen in any of the other sensors from PZS or PZN? Is the data from all the other sensors coherent in between? How can we know that P5 is not recording some unrelated disturbance? Section “Negative pore pressure and dilatancy”. A period with negative pore pressure is also seen at PZN. Why is it here not mentioned?	Yes the right reference is Fig S5, corrected in the new version. As mentioned in the manuscript (lines 131 and 132), the PZS-P5 sensor is positioned within a silty-sandy layer (Supplementary Fig. 9) potentially dilatant during shearing. In contrast, the other five sensors at PZS positioned within clayey sediments did not show any pore pressure perturbation during the monitoring period because clay will not behave in a similar way then silt under shearing.

21. L#143-146: More parameters of the simulation are needed: what is the magnitude estimated of the slow slip event to match the observation? What are the elastic and friction parameters? The depth is only constrained to be < 8km. It would be optimal if the depth of the slow slip event could be better constrained.	The slow slip event is equivalent to a M 5.1 earthquake, and is now mentioned in the methods section (see lines 420-422). This is a purely kinematic study, no frictional parameters were required and only the Poisson ratio is required for deformation calculations which are discussed in more detail in the methods section. With the inclusion of the magnitude estimation a shear modulus of 30 GPa was made which is mentioned. We agree that it would be optimal if we could better resolve the depth of the slow slip event – this is the reason why we tested a range of different slipping depth, however with only one geodetic observation it was not possible to better resolve this observation.
22. L#167, not only for subduction zones, but even in the Marmara region, see comment from L#44.	See also reviewer#2 comments and our reply.
23. L#170-171: But the inflation (dilatancy) is not seen in PZN, but only in one of the sensors from PZS, correct?	Yes. Dilatancy occurred only at the level of PZS-P5 sensor because of the presence of coarse material.
24. L# 180-182: The observed pressure drop 4 days before the 2013 earthquake is less than 1 kPa, therefore, it should lay in the range of the daily-fortnightly stress changes from tidal variations. Can such change just reflect a tidal variation in the sea level?	No this is not possible since we are measuring differential pore pressure.
Reviewer #2	
Comments	Reply
25. The manuscript presents a direct evidence of pore pressure changes associated with an aseismic tectonic transient (i.e., slow-slip event) along a branch of the North Anatolian Fault in the Sea of Marmara (Main Marmara Fault). The interpretation is based on the observation of the coupling between pore pressure changes, detected by near-fault piezometers, and tectonic deformation, recorded at a GPS station. I really enjoyed reading this paper, which is well written and organized, and the main results and concepts are nicely illustrated in the figures. Although the authors put together a very nice story, my only major concern is about the detection of the	We thank the reviewer for this positive comment.

aseismic slip signal at a single station as I will discuss below in my comments.	
26. As already mentioned, my main concern is about the tectonic transient signal (i.e., slow slip event) recorded from a single station. Records from single stations leave always open the possibility that the signal is due to a very local disturbance and make very difficult, and often not unambiguous, their interpretation. To overcome this limitation the authors, perform numerical modelling to better understand the origin/characteristics of the signal recorded at the GPS station TEKR. The modelling results indicate that the detected signal is compatible with an aseismic slip episode propagating at 0.5 km/day from E to W at shallow depths (0-8 km), along the Main Marmara Fault.	We agree with this comment regarding the limitation of an interpretation when it is based on a single measurement and a single signal. Unfortunately, the GPS station used is the only one close to the two events concerned by our monitoring period and we think that it is important to take advantage of this work to try to push towards a denser instrumentation network and maybe closer (offshore?) to the most active segment of the fault. This work should be seen as a first step towards more comprehensive analysis about fluid and seismicity in the area. We believe that the subject considered is so important for our community that even with partial data it is important to point out this coupling process between deep and surface processes and between fault activity and fluid pore pressure.
27. The possible slow slip event is recorded on the N-component while it produces no detectable deformation on the E-component (GPS station TEKR). I would imagine a slow-slip event originated along an E-W striking transform fault to produce a larger signal on the E-component than on the N-component as also the results from the modelling seem to show (Fig. 13-16S). I think the authors should discuss and explain this part a bit better (i.e., why the signal is visible on the N-component and not on the E-component?).	A possible explanation for this may be due to the use of an elastic isotropic, homogeneous model with uniform slip on the fault. In reality slip heterogeneity and the presence of normal faults may amplify north-south motion at the expense of east-west motion. There is now a discussion provide on this on lines 181-185
28. To my knowledge the vertical component of the GPS is the one affected by the largest errors and therefore the trickier to use. Is the +-1 cm displacement deficit above the noise level of the data? Which is the noise level of the data? Did the authors correct the data for precipitation records in the region or sea level variations? (e.g., can it be excluded that the signal is generated by a local peak in the rainfall or by sea level oscillations?). In general, I believe that a more detailed explanation of the processing of the geodetic data is needed to give the reader a better idea about the corrections applied (e.g., seasonal trend, steps from earthquakes, precipitations) to the data and if the noise level of the data are	A new paragraph about the processing GPS data is added to the new manuscript (see lines 76 to 92). An accuracy analysis of relative positions of permanent GPS stations in the Marmara Region carried out by Doğan has shown that the Root Mean Square Error (RMSE) is within 1 mm for the north-south, east-west components while it is between 2 to 3 mm for the up-down direction. The data is now corrected for the seasonal fluctuations. The interpretations remain valid with the new processed GPS data.

smaller than the amplitude of the detected signal(s).	
29. To strengthen the hypothesis of the tectonic origin of the geodetic signal at TEKR the authors could consider the following suggestions: - The repeater families reported in Schmittbuhl et al. (2016) cover the temporal interval during which the authors detect the tectonic transient. Is there an acceleration of the relative plate motion indicated by the repeaters? If yes, this could be a strong argument in support of the tectonic origin of the signal at the location suggested by the authors. If not, then the authors should try to explain why the acceleration of the relative plate motion is not evident in the slip rates inferred by the repeater sequences.	Schmittbuhl et al (2016) identify "nine long-lasting strike-slip seismic repeaters, in a 10 km region below the Central Basin at a depth > 8 km (except one repeater at 3.8 km depth) having a typical recurrence time of 8 months during the 2008–2015 period. They affirmed that "“The cumulative slip of the repeating sequence is compatible with the regional geodetic slip rate if they are assumed to be part of a large single asperity (10 km). The repeaters also exhibit short-term crises and are possibly related to “bursts of creep”. The duration (10 months) of the slipping event we observe is consistent with Schmittbuhl’s (8 months in average). This is now indicated in the manuscript. By comparing the cumulative slip calculated by Schmittbuhl et al (2016) to the geodetic data from TEKR (N-S trend) we can see that during the period englobing the 2013EQ and 2014EQ, the absence of any seismic activities within the shallow repeaters (i.e. at 3.8km and 8 km) fit well with the disturbance observed on the geodetic data (see figure below). During this period the deeper repeaters (i.e. > 8km) continue to slip seismically. One hypothesis could be related to the unlocking of the shallow repeaters for a short period related to slow slip and dilatancy. For example the dilatancy may change the behavior of failure mechanism by increasing the nucleation length required for unstable slip to a point where it slips a quasi-static manner (i.e. $L_c \propto 1/\sigma'$ where σ' is the effective normal stress on the fault , Rubin and Ampuero, 2005) However, at this stage the interpretation of this coincidence is more speculation than scientific demonstration. Therefore, we would like to avoid including this comparison in the main paper.
- How the recorded signal in this study compares with other aseismic slip signals	The best fitting thickness of the slipping patches we model is shallow (< 8 km) and is consistent with Yamamoto et al (2019);

detected along the North Anatolian Fault in terms of propagation velocities and depth intervals? (e.g., Aslan et al., 2019; Rousset et al., 2016).

Rousset et al (2016) and Aslan et al (2019) results. See also our detailed reply above (#9).

30. Fig. 1A: do the authors mean Fig. 2B (above the black arrow indicating the study region) or should it be Fig. 1B?

It is figure 1B showing the study area with PZN and PZS. Corrected.

31. Fig. 1B: coordinates are missing (same as Fig. 1A-C of the supplement).

Ok corrected.

32. Fig. 4: It could be useful to also indicate the location of the piezometers and indicate the temporal variation of the pore pressure together with the deformation observed at station TEKR (already in the figure).

It is now figure 5.
Ok piezometer locations are added. The temporal variation of PZS-P5 is already in Figure 5a. In the new version we added to the figure the data from PZN-P6.

Basically, to synthesize what the authors mention in ln. 111-113 the authors could use a different color to show/indicate: (1) TEKR displacement towards South and pore pressure increase at PZN-P6, and (2)

Those data are already indicated in figure 5a. Different colors are now used to differentiate between the different phases (displacement towards South and associated pore pressure

TEKR displacement towards North and pore pressure decrease.	and displacement towards north and the corresponding pore pressure).
33. Concerning the effect of pore pressure changes and fault valve behavior in regulating tectonic slip I think that there are two recent papers that are omitted in the references both in the introduction (ln. 40) and in the discussion (ln. 170). Gosselin et al. (2020) and Warren-Smith et al. (2019) provide seismological evidence for fault-valve behavior proposing it as the mechanisms controlling the genesis of slow-slip earthquakes. Probably they should be integrated in the manuscript.	Both references are accurate and are now added in the introduction paragraph.
34. I came across the paper of Proctor et al. (2020) about “direct evidence for fluid pressure, dilatancy, and compaction affecting slip in isolated faults”. How pore-pressure changes during slip dilatancy compare with the one presented in the Proctor et al. paper? This could be an interesting part to include in the discussions. In fact, the observations of the authors rely on a single occurrence which does not imply a repetitive occurrence of such behaviour, e.g., should we always expect a pore-pressure drop before the nucleation of an earthquake (ln. 119-121)? Laboratory experiments by reproducing multiple deformation cycles (e.g., Proctor et al., 2020) could help to address it.	A paragraph is added to mention the results of Proctor et al. and to point out the major role of pore fluid-sediment interactions in controlling and accompanying the process of fault slipping (lines 191 to 195). Concerning the drop of pore pressure before the nucleation of the earthquake (lines 119-121 in the old version), this was not considered as a major observation in our analysis because of the unicity of the event and none of the modeling results or the interpretation has focused on this event. In the new version the two lines concerning the drop of the pore pressure before the 2013EQ have been removed (see lines 149-151)
35. station TEKR is written both in capital and lowercase (e.g., ln. 147, ln. 159, Fig. 3-4), the authors may want to uniform it.	Ok correction made. TEKR is written in capital in the new manuscript.
36. ln. 114-115: “wet period” and “dry period” refer to the system (i.e., wet/dry conditions) or to the weather and therefore to the amount of precipitations? Needs to be clarified.	Wet period corresponds to high Precipitations, added in the text (line 144).
37. ln 158: “incontestably” is the most appropriate term to use in this case?	Ok replaced by probably
38. ln. 167: since there is evidence for creep bursts along the North Anatolian Fault it could make sense to refer also to such papers (Aslan et al., 2019; Rousset et al., 2016)	Considered see points #9 and #29
Reviewer #3	
Comments	Reply
39. I find the manuscript very interesting and of critical importance in shedding light	We thank the reviewer for this positive comment. The comment concerning the

on the physics of important hydraulic phenomena relating deep processes to near-surface and surface observables. However I find the presentation of the material somewhat confusing as I indicated in my doc file.	presentation of the material will be considered below by replying to the detailed comments.
40. The work is mainly targeting to understand the mechanisms through which near-surface observations of pore pressure and geodetic measurements are affected by the processes at depth. The relationship between the slow slip events and hydraulic phenomena have indeed been documented before (mostly for subduction zones), so in this context studying a well-documented faults zone such as the MMF is indeed very interesting.	We thank the reviewer again for this positive comment concerning the subject of the paper.
41. The authors detect a pore pressure transient preceding a seismic event and relate the polarity of the transient to the onland geodetic observations to conclude a, so to speak, teleconnection between both observables and the physical event through shear dilatancy which is basically the volume change observed in granular materials when they are subjected to shear deformations. In that respect, if shear dilatancy is indeed proven to be playing the role that the authors claim, then monitoring pore pressure transients continuously would indeed be very valuable to better understand seismicity.	Our data from FPZS-P5 indicate indeed the occurrence of dilatancy during a period fitting well with the signal perturbations recorded by PZ6 and the onshore geodetic data at TEKR.
42. I find the results very interesting, however I find the presentation of the manuscript confusing and not properly sequenced, forcing the reader to concentrate on several aspects of the phenomena simultaneously.	Thank you. For the presentation, see our replies below. The paper is now organized to follow the structure requested by the journal.
43. The effect of the deep tectonic processes in causing the surface elastic vertical displacements is discussed without a proper explanation of the geometry. A schematic figure to explain the geometry of the numerical model (together with a proper explanation of the boundary conditions) is necessary.	This is now done in the new version. See mainly the new Supplementary Fig. 12 and the new paragraph entitled: 3D displacement field in half-space linear elastic medium due to shear and tensile along the MMF
44. I would also be willing to see the numerical code used for both the elastic displacement and advection-diffusion.	The diffusion/advection code and input and output files are available on https://github.com/nsultan-2021/advection-diffusion

	The elastic displacement code has been made available at : https://github.com/s-murphy/StrikeSlipDef
45. Also missing is a supplement of the mathematics that is used for this model with at least some detail in the numerical method.	This was already done for the advection/diffusion code (page 11 in the old version). For the elastic displacement code, additional equations and explanation are now added to the new version in the paragraph “3D displacement field in half-space linear elastic medium due to shear and tensile along the MMF” in the methods section.
46. The formulation of the pressure transients in terms of the advection-diffusion equation is better discussed but the link between this model and the surface elastic phenomenon is “lost in translation”.	Modeling is done at two different scales: at the piezometer scale (<10 mbsf) by considering the fluid-flow in the porous medium and at the scale of the basin where the medium is considered as elastic (non-porous). A quantitative link between the two scales is not possible however the description of the physical phenomenon corresponding to the dilatancy, pore pressure increase/decrease fracturing and displacement are all linked together in the last paragraph of the paper and the figure 5. Taken all those observed and modelling results together, it was possible to draw a timeline of events that occurred after the 2013EQ indicating the way the aseismic creep affects the mud volcano activities, the pore pressure at the level of the piezometers as well as the 3D displacement field surrounding the MMF Figure 5 was modified by including different colors to differentiate between the different phases (displacement towards South and associated pore pressure from piezometers and displacement towards north and the corresponding measured pore pressure).
47. The slip scenarios discussed at the end of the text indeed shed light on the possible effect on the geodetically measurable observations but a scale analysis is missing to relate the results to quantitatively tie to the pressure modeling (and observations). A better structured text would definitely help this aspect. I find the approach logical but difficult to follow.	The new version of figure 5a shows the geodetic data, the modelling results and the pore pressure data. We believe that all those data together should help to clarify the described mechanism in Fig. 5b

48. Another confusing aspect is the fact that the temperature data is discussed in a very qualitative way. The way in which the temperature field is coupled with the ongoing pressure transients must be discussed using the basic thermodynamics of the granular system under consideration. Furthermore some statements were given without a proper explanation, especially in the discussion of demarcating the zones dominated either by diffusion or advection. For instance in the statement below: “Thermal gradients at PZS (Supplementary Fig. 7b) do not allow concluding about the thermal process controlling the temperature field while the hydraulic gradients at PZS show that only the sensor P5 at 6.28 mbsf is concerned by the pore pressure perturbations “ Here it is not clear what is meant by “thermal process”. It is also not clear why the gradients at the PZS do not allow whether or not these thermal processes control the temperature field. I am not asking the author to use a n additional energy equation to couple their existing numerical model to properly model the temperature field but I expect clearer explanations.	The interpretation of the temperature data is based on the shape of the temperature profile. For PZN, the thermal profile suggests that the temperature field at the level of the upper four sensors is in a permanent regime and primarily diffusion-controlled (quasi-constant gradient). This is because the linearity of the profile. For the deepest two sensors (quasi-constant temperature), the temperature is advection-controlled with a thermal gradient almost equal to zero. The idea is to check if the diffusion or advection is controlling the temperature profile. For PZS, the thermal data indicate a transient temperature regime without the possibility to conclude about the thermal process (advection or diffusion) controlling the temperature field. Indeed the non-linearity of the temperature profile could be the result of a transient diffusion of advection processes. This is added to the new version (see lines 303 to 307).
49. All in all this is a very interesting manuscript but a reorganization of the text with the issues mentioned above is necessary.	We thank the reviewer for this final positive comment.

REVIEWERS' COMMENTS

Reviewer #1 (Remarks to the Author):

The authors have thoroughly addressed my previously raised concerns as well as those raised by the other reviewers and it is suitable for publication in present form.

Reviewer #2 (Remarks to the Author):

Dear Editor,

hereby I send you my review of the revised version of the Sultan et al. paper entitled "Creep-dilatancy development at a transform plate boundary".

Please notice that the line number when it is mentioned refers to the annotated version of the manuscript "NCOMMS-21-23282-T-A_annotated.pdf."

The authors addressed the main comments and/or concerns of the reviewers and that has led to an improved version of the manuscript. The improvements mainly concern the description of the methods that is now more detailed. The authors also included new figures and/or modified some of them both in the main text and supplement that now help to better visualize concepts discussed in the paper.

The manuscript, as already mentioned in my initial review, addresses a scientifically relevant topic, and provides direct evidence of pore pressure changes associated to a slow slip event by combining geodetic and in-situ pore-pressure measured at piezometers. However, the limited amount of data leaves a bit of ambiguity in the obtained results as the same authors honestly state in some of the response to the reviewers e.g. "We believe that the subject considered is so important for our community that even with partial data it is important to point out this coupling process between deep and surface processes and between fault activity and fluid pore pressure." or even "we can just hypothesis that measured pore pressures at PZN-P6 and onshore geodetic perturbations have the same source."

In the first part of the review, I include some comments on the rebuttal letter, where the comment number is the same as the one reported from the authors, and then follow comments on the annotated version of the manuscript.

Comments to the Rebuttal letter:

Response to comment 13. I am a bit confused here. The reviewer mentions the Martinez-Garzon et al. (2019) paper in EPSL. The authors answer citing the Martinez-Garzon et al. (2019) paper in EPSL, however, the results the authors describe refer to Martinez-Garzon et al. (2019) in Tectonophysics. The reference should be corrected, or the text should be changed to reflect the results of Martinez-Garzon et al. (2019) in EPSL.

Martínez-Garzón, P., Ben-Zion, Y., Zaliapin, I., & Bohnhoff, M. (2019). Seismic clustering in the Sea of Marmara: Implications for monitoring earthquake processes. *Tectonophysics*, 768, 228176.

Martínez-Garzón, P., Bohnhoff, M., Mencin, D., Kwiatek, G., Dresen, G., Hodgkinson, K., ... & Kartal, R. F. (2019). Slow strain release along the eastern Marmara region offshore Istanbul in conjunction with enhanced local seismic moment release. *Earth and Planetary Science Letters*, 510, 209-218.

Response to comment 17: The authors write “A shallow localized process at the level of the piezometer is impossible to be detected by onshore GPS data and vice versa”. I think it could be nice to integrate this part also in the main text.

Response to comment 29. I think it could be worth to include also in the supplement of the paper to figure prepared to answer the comment from the reviewer. The agreement between seismological and geodetic observations could add more value to the paper.

Response to comment 33. I believe that the two papers are not cited in the exact context. The authors discuss the fault valve behavior and the relations between pore-fluid pressures and slow slip events which is where I would have expected to see both the papers cited. I do not find them relevant in the context where they appear now “Ln: 35-37”.

Comments to the manuscript.

The authors should double-check the numbering of the first ten references because there is no agreement between the annotated and non-annotated version of the manuscript.

Ln 71: two notable earthquakes occurred near-by. I think the term “near-by” is a bit too vague and the authors should be more specific.

Ln 81: with “observing session” do the author mean “observing period”?

Ln 82-84: I think it needs to be made clearer that the reported accuracy come from another study. e.g. the authors could start the sentence with: “a previous study ...”. Doğan (2007) uses different geodetic stations with respect to those used in this paper, so my question is if and to which extent the accuracy comparable? Furthermore, the authors mention that the accuracy of the GPS solutions is dependent on the observation periods, so to which duration of the observing period do the reported accuracy refer?

Ln 84-86: It is not clear to me why the authors mention “Additional processed data are available from NGL ... least squares method.” Is the sentence needed? Do the authors use these additional processed data?

Ln 177-178: the authors may want to specify that the reported results come from the “same” region.

Ln 182: MMR = MMF?

Ln 184: rephrase “explanation could” as “explanation could be”.

Ln 194: replace “fault” with faults.

Fig 5a-b: I am a bit confused with the “legend” on the top left of each panel. Why some of the lines/symbols do not have labels?

In the sections “Seabed amplitude and sub-seabed seismic features” and “Geodetic data” of the Methods I do not see a real description of the methods used in the analysis. I would either add a more extensive description of the methods or perhaps remove them.

Reviewer #3 (Remarks to the Author):

The revised manuscript, as in its present state is satisfactory and I am willing to let to to be published.

The authors thank the Editor and the referees for their constructive comments. In the following, we give a point-to-point reply (blue) to the referee comments.

The line number when it is mentioned refers to the NCOMMS-21-23282B_annotated.pdf.

Editor - Nature Communications	
Comments	Reply
referee #2 has concerns about the fact that your study is based on data from a single GPS station only. After discussing this with referee #1 and my team manager, we are happy to move forward, since it seems there is no possibility to get additional data. However, in addition to revising your manuscript towards the remaining comments of the referees, we would like to ask you to include a clear caveat in your abstract and discussion towards the fact that all data comes from a single GPS station only. This should be clearly conveyed to the reader. In the abstract, you could i.e. add something like: Here, we use offshore in-situ sediment pore-pressure acquired in the proximity of the active offshore Main Marmara Fault and onshore geodetic time-series data set from a single GPS station to demonstrate the pore-pressure/deformation coupling during a 10-month slow-slip event.	Thank you for your relevant assessment. The abstract was modified following your suggestion.
At the same time we ask that you edit your manuscript to comply with our policies and formatting requirements and to maximise the accessibility and therefore the impact of your work.	Done as requested
Please see the attached document(s), listing a number of points that must be addressed. Failure to comply with our editorial requests will cause delays in accepting your manuscript.	All the requests made were fulfilled.
Reviewer #1	
Comments	Reply
The authors have thoroughly addressed my previously raised concerns as well as those raised by the other reviewers and it is suitable for publication in present form.	The authors thank the referee for this positive feedback
Reviewer #2	
Comments	Reply
The manuscript, as already mentioned in my initial review, addresses a scientifically relevant topic, and provides direct evidence of pore pressure changes associated to a slow slip event	We thank the reviewer for this positive comment.

by combining geodetic and in-situ pore-pressure measured at piezometers.	
However, the limited amount of data leaves a bit of ambiguity in the obtained results as the same authors honestly state in some of the response to the reviewers e.g. “We believe that the subject considered is so important for our community that even with partial data it is important to point out this coupling process between deep and surface processes and between fault activity and fluid pore pressure.” or even “we can just hypothesis that measured pore pressures at PZN-P6 and onshore geodetic perturbations have the same source.”	We agree with this comment but as mentioned previously, the GPS station used is the only one close to the two earthquake events concerned by our monitoring period. As requested by the editor a sentence is added in the abstract clearly stating that only one geodetic station was used in this work (lines 21-22).
Response to comment 13. I am a bit confused here. The reviewer mentions the Martinez-Garzon et al. (2019) paper in EPSL. The authors answer citing the Martinez-Garzon et al. (2019) paper in EPSL, however, the results the authors describe refer to Martinez-Garzon et al. (2019) in Tectonophysics. The reference should be corrected, or the text should be changed to reflect the results of Martinez-Garzon et al. (2019) in EPSL. Martínez-Garzón, P., Ben-Zion, Y., Zaliapin, I., & Bohnhoff, M. (2019). Seismic clustering in the Sea of Marmara: Implications for monitoring earthquake processes. Tectonophysics, 768, 228176. Martínez-Garzón, P., Bohnhoff, M., Mencin, D., Kwiatek, G., Dresen, G., Hodgkinson, K., ... & Kartal, R. F. (2019). Slow strain release along the eastern Marmara region offshore Istanbul in conjunction with enhanced local seismic moment release. Earth and Planetary Science Letters, 510, 209-218.	Reference corrected.
Response to comment 17: The authors write “A shallow localized process at the level of the piezometer is impossible to be detected by onshore GPS data and vice versa”. I think it could be nice to integrate this part also in the main text.	The sentence is added to the main text (lines 138 to 140).
Response to comment 29. I think it could be worth to include also in the supplement of the paper to figure prepared to answer the comment from the reviewer. The agreement between seismological and geodetic observations could add more value to the paper.	As explained in our previous response letter to reviewers, we believe that the interpretation concerning the agreement between seismological and geodetic observations is not scientifically strong enough to be included in the paper. However, the files including the reviewers comments and our responses are

	now available online and can be accessed by readers.
Response to comment 33. I believe that the two papers are not cited in the exact context. The authors discuss the fault valve behavior and the relations between pore-fluid pressures and slow slip events which is where I would have expected to see both the papers cited. I do not find them relevant in the context where they appear now “Ln: 35-37”.	Corrected as suggested. The two references are mentioned at the end of the following sentence “in situ observations from the seafloor along the subducting plate interfaces have led to the hypothesis of a causal relationship between SSEs and changes in fluid activities at the fault zone”
The authors should double-check the numbering of the first ten references because there is no agreement between the annotated and non-annotated version of the manuscript.	The numbering is correct in the non-annotated version. The numbering is done automatically using endNote and will only be displayed correctly when accepting the corrections
Ln 71: two notable earthquakes occurred near-by. I think the term “near-by” is a bit too vague and the authors should be more specific.	Corrected as suggested. Near-by is replaced by the epicentral distances from the piezometers (lines 71-72).
Ln 81: with “observing session” do the author mean “observing period”?	Corrected as suggested. Session is replaced by period.
Ln 82-84: I think it needs to be made clearer that the reported accuracy come from another study. e.g. the authors could start the sentence with: “a previous study ...”. Doğan (2007) uses different geodetic stations with respect to those used in this paper, so my question is if and to which extent the accuracy comparable? Furthermore, the authors mention that the accuracy of the GPS solutions is dependent on the observation periods, so to which duration of the observing period do the reported accuracy refer?	Corrected as suggested about mentioning “a previous study”. The study carried out by Doğan (2007) concerns indeed some stations of the Marmara Continuous GPS Network (MAGNET). However, this author concludes that “The results of this investigation show that highly accurate positional coordinates can be obtained using MAGNET in the Marmara region”. Therefore, we trust that the accuracy of the analyzed data from TEKR is comparable to other stations from MAGNET.
Ln 84-86: It is not clear to me why the authors mention “Additional processed data are available from NGL ... least squares method.” Is the sentence needed? Do the authors use these additional processed data?	We agree with this comment. Sentence deleted.
Ln 177-178: the authors may want to specify that the reported results come from the “same” region.	Corrected as suggested.
Ln 182: MMR = MMF?	Corrected as suggested. MMR replaced by MMF
Ln 184: rephrase “explanation could” as “explanation could be”.	Corrected as suggested
Ln 194: replace “fault” with faults.	Corrected as suggested
Fig 5a-b: I am a bit confused with the “legend” on the top left of each panel. Why some of the lines/symbols do not have labels?	Legend completed.
In the sections “Seabed amplitude and sub-seabed seismic features” and “Geodetic data”	We prefer to maintain these two paragraphs as they allow us to clearly mention the origin of

of the Methods I do not see a real description of the methods used in the analysis. I would either add a more extensive description of the methods or perhaps remove them.	the data used and to refer to two essential figures in the supplementary materials
Reviewer #3	
Comments	Reply
The revised manuscript, as in its present state is satisfactory and I am willing to let to to be published	The authors thank the referee.